# CP4D: Compositional Physics-aware 4D Scene Generation

## Abstract

4D generation (*i.e.*, dynamic 3D generation) has recently emerged as a rapidly growing research frontier due to its powerful spatiotemporal modeling capabilities. However, despite notable advances, existing approaches typically fail to capture the underlying physical principles, producing results that are both physically inconsistent and visually implausible. To overcome this limitation, we present CP4D, a novel paradigm for photorealistic 4D scene synthesis with faithful adherence to complex physical dynamics. Drawing inspiration from the compositional nature of real-world scenes, where immutable static backgrounds coexist with dynamic, physically plausible foregrounds, CP4D reformulates 4D generation as the integration of a static 3D environment with physically grounded dynamic objects. On this basis, our framework follows a three-stage pipeline: **1)** Firstly, we leverage pre-trained expert models to generate high-fidelity 3D representations of the environment and foreground objects respectively. **2)** Subsequently, to produce physically plausible trajectories and realistic interactions for these objects, we propose a hybrid motion synthesis strategy that integrates priors from physical simulators with the common sense embedded in video diffusion models. **3)** Finally, we develop an automated composition mechanism that seamlessly fuses the static environment and dynamic objects into coherent, physically consistent 4D scenes. Extensive experiments demonstrate that CP4D can generate explorable and interactive 4D scenes with high visual fidelity, strong physical plausibility, and fine-grained controllability, significantly outperforming existing methods. The anonymous project page: `https://anonymous.4open.science/w/CP4D/`.

## 1 Introduction

Empowered by recent progress in generative models (Ho et al., 2020; Song et al., 2020) and large-scale data available, 4D generation (*i.e.*, dynamic 3D generation) (Ren et al., 2023; Xie et al., 2024b; YU et al., 2025; Ma et al., 2025) has emerged as a prominent research focus. Through joint modeling of spatial structure and temporal dynamics, 4D generation enables the synthesis of realistic and coherent 4D scenes, holding great promise for a wide range of applications such as AR/VR (Li et al., 2024a), robotics (Liu et al., 2025a), and world models (Chen et al., 2025b).

Existing approaches for 4D generation can be broadly divided into two categories. The first class of methods exploits priors distilled from pre-trained video or 3D generative models (Bahmani et al., 2024b;a; Jiang et al., 2023; Zeng et al., 2024), employing them as auxiliary supervisory signals to constrain the generation process and improve fidelity. In contrast, the second class follows a data-driven paradigm (Xie et al., 2024b; Ren et al., 2024; Liang et al., 2024; Bai et al., 2025), where cross-view videos are directly synthesized as intermediate proxies and subsequently transformed into full 4D content through classical reconstruction pipelines. While producing seemingly plausible results, these approaches typically lack an explicit characterization of the underlying physical principles. As a consequence, the generated content often suffers from physical inconsistencies and visual artifacts, leading to scenes that deviate from realistic dynamics.

To mitigate this issue, inspired by the compositional nature of real-world scenes (Xu et al., 2024; Zhu et al., 2024), where static backgrounds co-exist with physically plausible dynamic foregrounds, we reformulate 4D scene generation as the integration of a static 3D environment with physically

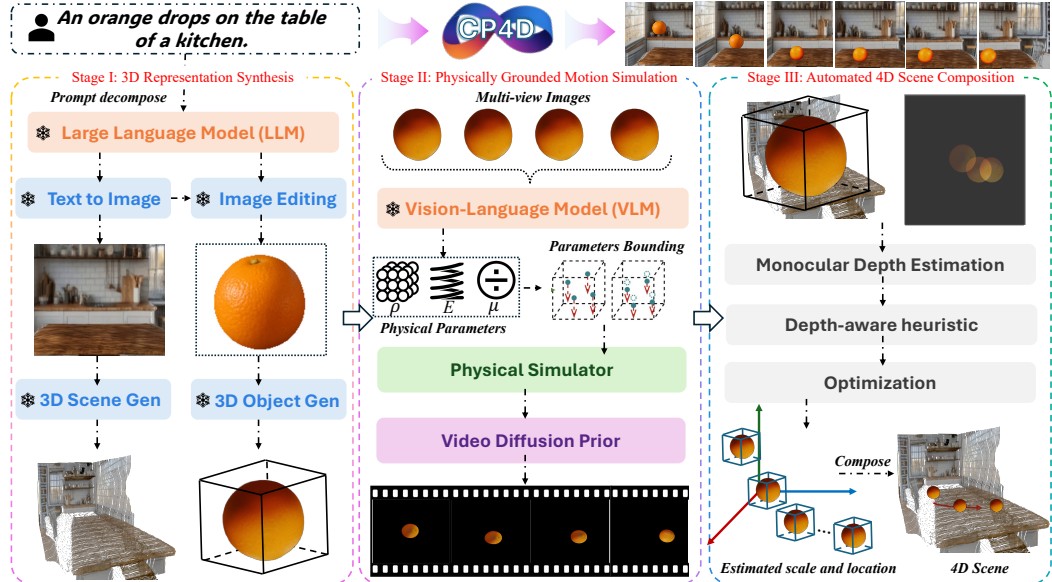

Figure 1: **Pipeline of CP4D.** Given a textual prompt, CP4D constructs a physically faithful 4D scene via a three-stage pipeline: 1) synthesizing 3D representations for both foreground objects and background environments (Sec. 4.1), 2) simulating foareground motions with physical grounding to ensure realistic dynamics (Sec. 4.2), and 3) automatically composing the foregrounds and background into a coherent and visually plausible 4D scene (Sec. 4.3).

grounded dynamic 3D objects. Building upon this formulation, there arise three key technical challenges: **1)** How to construct plausible 3D representations of background environments and foreground objects that conform to user-specified instructions? **2)** How to model the motion dynamics of foreground objects that encompass physically plausible trajectories and realistic interactions? **3)** How to seamlessly compose the generated dynamic foregrounds with the static background into a consistent 4D scene?

To tackle these challenges, in this paper we introduce CP4D, a novel paradigm for photorealistic 4D scene generation with faithful adherence to complex physical dynamics. Specifically, as shown in Fig. 1, CP4D follows a three-stage pipeline: **1)** Firstly, given a textual prompt, we first synthesize a background image using a text-to-image generative model, after which an image editing model, conditioned on this background, is employed to generate foregrounds that are visually compatible with it. Both the background and the foregrounds are then reconstructed into their respective 3D representations using pre-trained expert models. In contrast to the naive baseline that independently applies text-to-3D models to each component, our approach enforces stylistic coherence across background and foreground, thereby mitigating implausible artifacts such as realistic environments juxtaposed with cartoon-like objects. **2)** Secondly, to endow foreground objects with physically plausible trajectories and realistic interactions, we introduce a hybrid motion synthesis strategy. In particular, we first leverage physical simulators to produce coarse object trajectories that comply with fundamental physical laws. These initial dynamics are subsequently refined using the commonsense knowledge embedded in video generative models, thereby enhancing inter-object interactions and yielding motion that is both more realistic and visually convincing. **3)** Thirdly, to seamlessly fuse the dynamic foregrounds with the static background into a unified 4D scene, we develop an automated composition mechanism. By leveraging monocular depth estimation and a depth-aware heuristic rule, this mechanism first estimates the relative spatial attributes of foreground objects (*e.g.*, positions and scales) within the background, which are subsequently calibrated through optimization to ensure coherent integration and visually compelling compositions.

Notably, owing to its compositional design, CP4D not only enables the synthesis of 4D scenes that faithfully comply with physical laws, but also provides strong interactive controllability. In particular, users are afforded the flexibility to edit different scene elements, such as foreground objects, background environments, and motion trajectories, thus facilitating diverse 4D generation.

In summary, our key contributions can be concluded as follows:

- We present CP4D, a novel compositional framework designed to generate photorealistic 4D scenes with accurate adherence to complex physical dynamics.

- We propose a hybrid motion synthesis strategy that integrates physical priors from differentiable simulators with commonsense knowledge from video generative models, yielding physically plausible trajectories and realistic interactions.

- We develop an automated composition mechanism that harmoniously fuses dynamic foregrounds with the static background, producing a coherent and visually compelling 4D scene.

- Extensive experiments demonstrate that CP4D is capable of synthesizing explorable and interactive 4D scenes characterized by high visual fidelity, robust physical realism, and precise controllability, consistently outperforming prior methods.

## 2 RELATED WORKS

### 2.1 4D GENERATION

Generating 4D assets from textual prompts has drawn growing attention owing to its wide-ranging applications in AR, VR (Wang et al., 2025), and spatial intelligence. Early approaches (Jiang et al., 2023; Zhu et al., 2025; Li et al., 2024b; Zeng et al., 2024; Gao et al., 2024) towards this goal predominantly relied on distilling knowledge from pre-trained generative models to guide the generation process. For instance, DreamGaussian4D (Ren et al., 2023) pioneered the use of SDS (Poole et al., 2022) in the 4D generation domain, demonstrating the capability to produce realistic 4D objects conditioned on text prompts. Consistent4D (Jiang et al., 2023) realized video-to-4D generation by integrating SDS with dynamic NeRF (Park et al., 2021), and further employed a video enhancer to improve the quality of the synthesized 4D assets. Recently, the availability of large-scale datasets (Deitke et al., 2023; Nan et al., 2024) has enabled methods that directly train feed-forward video diffusion models to synthesize multi-view videos (Xie et al., 2024b; Ren et al., 2024; YU et al., 2025; Bai et al., 2025; He et al., 2024; Namekata et al., 2024), which are subsequently reconstructed into 4D scenes using standard reconstruction techniques (Wu et al., 2024). However, despite their ability to produce seemingly plausible results, these approaches generally overlook the explicit characterization of underlying physical dynamics. Consequently, the generated content often exhibits physically inconsistent behaviors and visual artifacts. In contrast, we present CP4D, a physics-aware framework for text-driven 4D scene generation, delivering photorealistic quality, reliable physical consistency, and precise generation control.

### 2.2 PHYSICS-BASED SIMULATION

Given an initial 3D representation (*e.g.*, 3D gaussian splatting (Kerbl et al., 2023)), recent works (Xie et al., 2024a) have explored the use of physical solvers, such as the Material Point Method (MPM) (Hu et al., 2018; Jiang et al., 2017), to update the state of Gaussian primitives under external forces at different timestamps. To automate the specification of material parameters, multimodal large language models (MLLMs) have been employed to infer properties such as density, Young's modulus, and Poisson's ratio (Zhao et al., 2024; Mao et al., 2025). Complementary to this, other approaches (Huang et al., 2025; Liu et al., 2024a; Lin et al., 2025; Liu et al., 2025b) exploit implicit physical regularities in video diffusion models by incorporating Score Distillation Sampling (SDS) (Poole et al., 2022) to refine these preliminary estimates. While the above methods assume access to well-defined 3D representations, more recent works (Lin et al., 2024a;b; Chen et al., 2025a; Tan et al., 2024; Liu et al., 2024b) aim to generate physics-driven videos directly from a single image. These methods first generate a full 3D representation using image-to-3D models (either mesh-based (Chen et al., 2025a) or Gaussian-based (Lin et al., 2024a;b; Tan et al., 2024)) before applying physical simulations as described above. However, existing solutions remain limited: they typically handle only elastic or rigid bodies, lack support for realistic multi-material Gao et al. (2025) and multi-object interactions, and often employ either 2D backgrounds or 3D environments with fixed viewpoints, restricting the ability to render consistent novel views.

## 3 PRELIMINARIES: SCORE DISTILLATION SAMPLING

Score Distillation Sampling (SDS) (Poole et al., 2022) is a widely used technique for optimizing a differentiable generator $g(\theta)$ under the guidance of a pre-trained diffusion model. Its core idea is to exploit the score function of the diffusion model to supply gradient that steer the generator's outputs towards alignment with a target text prompt, without the need for explicit likelihood computation.

Formally, let $\epsilon_\phi(\cdot, \mathbf{T}, \zeta)$ denote the denoiser of a pre-trained text-conditioned diffusion model parameterized by $\phi$, with timestep $\zeta$ and text prompt $\mathbf{T}$, the SDS gradient is given by:

$$\nabla_\theta \mathcal{L}_{\text{SDS}} = \mathbb{E}_{\epsilon, \zeta} \left[ \omega(\zeta) \left( \epsilon_\phi(g(\theta), \mathbf{T}, \zeta) - \epsilon \right) \frac{\partial g(\theta)}{\partial \theta} \right], \tag{1}$$

where $g(\theta)$ denotes the generator's output (e.g., a rendered video), $\epsilon$ is Gaussian noise sampled at timestep $\zeta$, $\omega(\zeta)$ is a weighting function, and $\theta$ are the learnable parameters of the generator.

## 4 METHODOLOGY

**Overview.** Given a textual prompt $\mathbf{T}$, our objective is to synthesize a 4D scene that faithfully adheres to complex physical dynamics while supporting flexible viewpoint changes. To this end, we adopt a compositional formulation grounded in the nature of real-world scenes (i.e., static backgrounds coexisting with physically governed, dynamic foregrounds), and cast 4D generation as the integration of a static 3D environment with physically grounded dynamic objects.

To achieve this goal, we introduce a three-stage pipeline. To begin with, we leverage pre-trained expert models to construct plausible 3D representations for both the background environment and the foreground objects (Sec. 4.1). Subsequently, we propose a hybrid motion synthesis strategy utilizing physical simulators and video generative models to produce foreground motions with physical consistency and realistic interactions (Sec. 4.2). Finally, we develop an automated composition mechanism that seamlessly integrates the generated background and foreground into a coherent 4D scene (Sec. 4.3).

### 4.1 STAGE I: BACKGROUND–FOREGROUND 3D REPRESENTATION SYNTHESIS

To achieve text-guided compositional physics-aware 4D scene generation (CP4D), constructing plausible 3D representations of both the background environment and the foreground objects constitutes an essential prerequisite, providing the foundation for subsequent motion modeling and scene composition. To this end, we first invoke a large language model (e.g., GPT-4o (Achiam et al., 2023)) to decompose the input textual prompt $\mathbf{T}$ into two sub-prompts (i.e., $\mathbf{T} = \{\mathbf{T}_b, \mathbf{T}_f\}$), each describing the background and foreground to be generated.

Subsequently, to obtain the corresponding 3D representations of the background and foreground, one intuitive approach is to independently apply pretrained text-to-3D generative models. However, such a straightforward strategy typically yields implausible outcomes, e.g., generating a realistic background paired with cartoon-like foregrounds, which in turn undermines the coherence and overall quality of the synthesized 4D scene.

To overcome this limitation, we adopt a simple yet effective strategy for 3D representation synthesis. Specifically, we first synthesize a background image $\mathbf{I}_b$ from the input prompt $\mathbf{T}_b$ using a text-to-image generative model $\mathbf{F}_{t2i}$. Next, conditioned on $\mathbf{I}_b$ and $\mathbf{T}_f$, we employ an image editing model $\mathbf{F}_{edit}$ to generate a composite image $\mathbf{I}_{b,f}$ that simultaneously contains both the background and foreground in a visually coherent manner. We then apply an image segmentation model $\mathbf{F}_{seg}$ to $\mathbf{I}_{b,f}$ to isolate the foreground region $\mathbf{M}_f$ ($\mathbf{M}_f = 1$ corresponds to foreground pixels and $\mathbf{M}_f = 0$ to background pixels), yielding the foreground image $\mathbf{I}_f$. Finally, with the harmonized background image $\mathbf{I}_b$ and foreground image $\mathbf{I}_f$, we leverage pretrained image-to-3D generative models $\mathbf{F}_{3d}^b$ and $\mathbf{F}_{3d}^f$ to construct their respective 3D representations. The overall pipeline can be formally expressed as follows:

$$\mathbf{G}_b = \mathbf{F}_{3d}^b(\mathbf{I}_b), \ \mathbf{G}_f = \mathbf{F}_{3d}^f(\mathbf{I}_f),$$
$$\mathbf{I}_b = \mathbf{F}_{t2i}(\mathbf{T}_b), \ \mathbf{I}_{b,f} = \mathbf{F}_{edit}(\mathbf{I}_b, \mathbf{T}_f), \ \{\mathbf{I}_f, \mathbf{M}_f\} = \mathbf{F}_{seg}(\mathbf{I}_{b,f}), \tag{2}$$

where $\mathbf{G}_b$ and $\mathbf{G}_f$ denote the 3D representations of the background and foreground, instantiated using *3D gaussian splatting*. For clarity, we use $\mathbf{G}_f$ as a unified notation to represent the foreground representation, which may correspond to either a single object or multiple different objects.

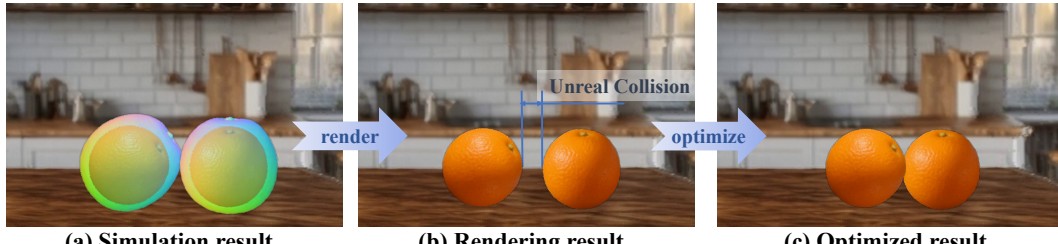

(a) Simulation result.     (b) Rendering result.     (c) Optimized result.

Figure 2: **(a)** Limited numerical precision in the physics solver leads to erroneous estimation of foreground geometry. **(b)** As a result, the solver reports collisions that are not visually manifested, producing spurious intersections. **(c)** Our method eliminates these inconsistencies, enabling interactions that are both visually coherent and physically faithful.

## 4.2 STAGE II: PHYSICALLY GROUNDED MOTION SIMULATION

Given the generated $\mathbf{G}_f$, the second stage aims to endow the foreground objects with motions that are both physically consistent and visually realistic. To this end, we adopt a hybrid motion synthesis framework: physical simulators are first employed to generate trajectories constrained by fundamental physical laws, which are subsequently refined using the commonsense priors embedded in video generative models. This design ensures that the resulting motions remain faithful to physics while exhibiting naturalistic interactions.

**Physical simulator-based motion synthesis.** To simulate the dynamics of $\mathbf{G}_f$ conditioned on the textual description $\mathbf{T}_f$, we begin by leveraging vision-language models (VLMs) to infer essential physical attributes of the objects, including material properties (*e.g.*, Young's modulus $E$, Poisson's ratio $\mu$, and density $\rho$) and external forces $\mathbf{Q}$. These inferred parameters provide the initialization required for physically grounded motion simulation. More details are provided in Appendix B.

We then employ heterogeneous physical solvers $\Phi$ to simulate object dynamics. Specifically, elastic or flexible objects are handled using an MPM solver $\Phi_{mpm}$, rigid objects are modeled with a dedicated rigid-body solver $\Phi_{rigid}$, while fluid objects are simulated with a Position-Base-Dynamic (PBD) solver $\Phi_{fluid}$ (More details are provided in Appendix C). Initialized with the estimated material parameters $\Theta = \{\rho, E, \mu\}$ and external forces $\mathbf{Q}$, the solvers evolve the foreground into deformed 3D representations $\mathbf{G}_f^t$ over time $t$, which can be expressed as:

$$\mathbf{G}_f^t = \Phi(\mathbf{G}_f, \mathbf{Q}, \Theta, t). \tag{3}$$

**Video generative model-based refinement.** Although Eq. 3 produces motions that are broadly consistent with physical principles, two critical limitations persist. **1)** As VLMs are not explicitly trained on physics-oriented datasets, the inferred material parameters, while generally reasonable, often lack the numerical accuracy required to reflect precise physical behavior. **2)** As shown in Fig. 2, physics solvers generally rely on grid-based approximations of $\mathbf{G}_f$ to model interactions such as collisions. However, the limited fidelity of these approximations often fails to capture the intricate geometry of the underlying 3D structures, leading to perceptually implausible outcomes, *e.g.*, collisions may be registered between objects despite no apparent contact in the rendered scene.

To mitigate these issues, we resort to commonsense knowledge embedded in video diffusion models. Specifically, to solve the first problem, we employ the SDS loss to optimize the estimated physical parameters $\Theta$, which is denoted as follows:

$$\nabla_\Theta \mathcal{L}_{\text{SDS}} = \mathbb{E}_{\epsilon, \zeta} \left[ \omega(\zeta)(\hat{\epsilon}_\psi(V; \mathbf{T}_f; \zeta) - \epsilon)\frac{\partial V}{\partial \Theta} \right], \tag{4}$$

where $V$ denotes the rendered video based on $\mathbf{G}_f^t$, $\hat{\epsilon}_\psi$ represents the predicted noise using pre-trained video diffusion model $\psi$, $\omega(\zeta)$ is a weighting function over the diffusion timestep $\zeta$.

To alleviate the second issue, namely the inaccuracies introduced by coarse grid-based approximations during inter-object interactions, we similarly employ SDS-based optimization. Specifically, assuming $\mathbf{G}_f$ comprises $K$ individual objects, $\{\mathbf{G}_{f_i}\}_{i=1}^K$, we assign to each object a learnable global displacement variable $\Delta\Gamma_i$, which adjust their relative positions. These displacement variables are

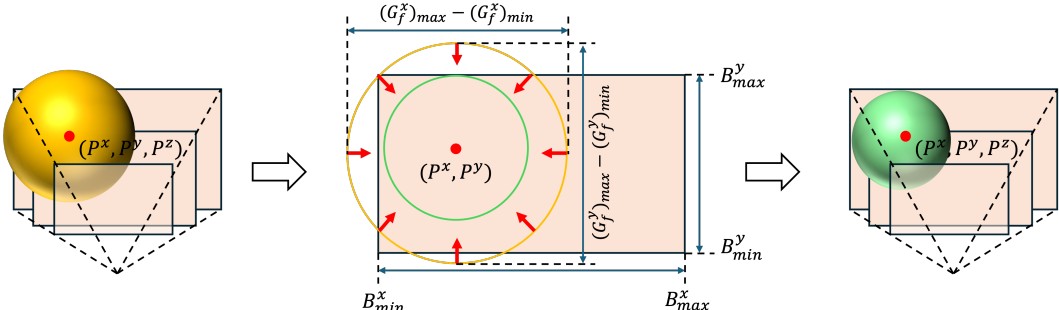

Figure 3: Illustration of the depth-aware heuristic for initializing the scale $S$. **Left:** the foreground $\mathbf{G}_f$ is independently generated and may exceed the camera frustum at depth $P^z$. **Middle:** to ensure full visibility, its projected extent in the $x$–$y$ plane is constrained by the frustum bounds $(B^x\text{min}, B^x_{\max}, B^y_{\min}, B^y_{\max})$, yielding the maximum feasible scale $S$. **Right:** applying this initialized $S$ guarantees that $\mathbf{G}_f$ remains entirely contained within the frustum of the reference view.

optimized via SDS supervision to ensure that the rendered video adheres to the textual prompt $\mathbf{T}_f$ while exhibiting interaction patterns aligned with human perceptual priors, which is formulated as follows:

$$\nabla_{\Delta\Gamma} \mathcal{L}_{\text{SDS}} = \mathbb{E}_{\epsilon,\zeta} \left[ \omega(\zeta) \big( \hat{\epsilon}_\psi(V_{\Delta\Gamma}; \mathbf{T}_f, \zeta) - \epsilon \big) \frac{\partial V_{\Delta\Gamma}}{\partial \Delta\Gamma} \right], \tag{5}$$

where $V_{\Delta\Gamma}$ denotes the rendered video after applying displacements $\Delta\Gamma$.

### 4.3 STAGE III: AUTOMATED 4D SCENE COMPOSITION

After obtaining physically grounded motions of the foreground object(s) $\mathbf{G}_f$, our next goal is to fuse them with the background $\mathbf{G}_b$ into a coherent 4D scene. To this end, we introduce an automated scene composition mechanism that estimates the relative spatial attributes of $\mathbf{G}_f$ (*e.g.*, its position and scale) with respect to $\mathbf{G}_b$ using monocular depth cues and heuristic priors, and further refines them through optimization to ensure both geometric consistency and visual plausibility. A detailed illustration is provided below.

**Relative spatial attributes initialization.** Since $\mathbf{G}_b$ and $\mathbf{G}_f$ are generated independently by different pre-trained expert models, their 3D representations lie in distinct coordinate spaces, making direct integration infeasible. Therefore, to reasonably place $\mathbf{G}_f$ into $\mathbf{G}_b$ with correct size and location, we propose to transform $\mathbf{G}_f$ into an aligned representation $\mathbf{G}_f^*$ using the following equation:

$$\mathbf{G}_f^* = S \times \mathbf{G}_f + P, \tag{6}$$

where $S \in \mathbb{R}^+$ denotes the relative scale and $P = (P^x, P^y, P^z) \in \mathbb{R}^3$ the relative translation. For clarity, we simplify $\mathbf{G}_f$ as $\mathbf{G}_f = (\mathbf{G}_f^x, \mathbf{G}_f^y, \mathbf{G}_f^z) \in \mathbb{R}^{U \times 3}$, considering only the transformation of its $U$ spatial coordinates.

Subsequently, to estimate the translation parameter $P$ (*i.e.*, the spatial location of $\mathbf{G}_f$ within $\mathbf{G}_b$), we employ a monocular depth estimator $\mathbf{F}_{depth}$ on the composite image $\mathbf{I}_{b,f}$ (as defined in Eq. 2) to recover a dense depth map of the scene. Guided by the foreground mask $\mathbf{M}_f$, depth values associated with the target region are isolated, from which the centroid depth of the foreground object is derived. This depth estimate is further back-projected into 3D space, providing an initialization of the foreground position $P$ in the coordinate frame of $\mathbf{G}_b$, which can be formulated as follows:

$$(P^x, P^y, P^z) = \Phi(\mathbf{D}_{b,f}[(M_f = 1)_{\text{cen}}]), \ \mathbf{D}_{b,f} = \mathbf{F}_{depth}(\mathbf{I}_{b,f}), \tag{7}$$

where $\mathbf{D}_{b,f}$ denotes the depth map estimated from the composite image $\mathbf{I}_{b,f}$, $(M_f = 1)_{\text{cen}}$ indicates the centroid pixel of the segmented foreground region, $\Phi(\cdot)$ represents the back-projection function that maps a 2D pixel into 3D space based on its depth value. Notably, since we unify the world coordinate system of the background with the camera coordinate system, the $z$-coordinate of $P$ (*i.e.*, $P^z$) is directly equal to the corresponding depth value $\mathbf{D}_{b,f}[(M_f = 1)_{\text{cen}}]$.

For scale estimation, *i.e.*, determining the size of $\mathbf{G}_f$ within $\mathbf{G}_b$, we employ a depth-aware heuristic. The key insight is that, under the reference viewpoint corresponding to $\mathbf{I}_{b,f}$, the foreground object

should be entirely visible within the image plane. This implies that, ***in 3D space, $\mathbf{G}_f$ must be fully contained within the camera frustum of the reference view.*** Given the estimated depth $P^z$, as shown in Fig. 3, the scale $S$ is constrained such that all points of $\mathbf{G}_f$ fall within the valid frustum slice at depth $P^z$, *i.e.*, their $x$- and $y$-coordinates remain bounded by the image-plane limits defined at that depth. Accordingly, we initialize $S$ as the maximum feasible scale that satisfies these geometric bounds, which is formulated as follows:

$$ S = \frac{\min(\min(P^x - B_{\min}^x, \, B_{\max}^x - P^x), \, \min(P^y - B_{\min}^y, \, B_{\max}^y - P^y))}{\max\left((\mathbf{G}_f^x)_{\max} - (\mathbf{G}_f^x)_{\min}, \, (\mathbf{G}_f^y)_{\max} - (\mathbf{G}_f^y)_{\min}\right)/2}, \tag{8} $$

where $B_{\min}^x, B_{\max}^x, B_{\min}^y, B_{\max}^y$ denote the horizontal and vertical boundaries of the camera frustum at depth $P^z$, $(\mathbf{G}_f^x)_{\max}, (\mathbf{G}_f^x)_{\min}, (\mathbf{G}_f^y)_{\max}, (\mathbf{G}_f^y)_{\min}$ represent the maximum and minimum $x$- and $y$-coordinates of the original foreground representation $\mathbf{G}_f$, respectively.

**Optimization-based refinement.** After obtaining the initial estimates of $P$ and $S$, we further refine them to improve perceptual fidelity. The objective is to ensure that the rendered reference view of the composed scene closely aligns with the composite image $\mathbf{I}_{b,f}$. Accordingly, we optimize $P$ and $S$ by minimizing the discrepancy between the rendered image $\hat{\mathbf{I}}_{b,f}(P, S)$ and $\mathbf{I}_{b,f}$, formulated as:

$$ (P, S) = \arg\min_{P, S} \left\|\hat{\mathbf{I}}_{b,f}(P, S) - \mathbf{I}_{b,f}\right\|_2^2. \tag{9} $$

Notably, our experiments reveal that simultaneously optimizing $S$ and $P$ introduces substantial ambiguity, often leading to suboptimal local minima. To address this, we employ a sequential refinement strategy: first optimizing the scale $S$, followed by refining the translation $P$. This progressive scheme significantly reduces uncertainty and consistently yields more robust and reliable composition results.

## 5 EXPERIMENTS

### 5.1 EXPERIMENTAL SETUPS

**Implementation details.** We curate a dataset of 17 examples for evaluation, where each instance consists of a foreground prompt $\mathbf{T}_f$ and a background prompt $\mathbf{T}_b$. Qwen-Image (Wu et al., 2025) is employed to generate the background image $\mathbf{I}_b$ from $\mathbf{T}_b$, and Qwen-Image-Edit is further applied to synthesize the composite image $\mathbf{I}_{b,f}$. The foreground mask $\mathbf{M}_f$ is extracted from $\mathbf{I}_{b,f}$ using SAM (Kirillov et al., 2023), and its depth map is estimated with Depth Anything (Yang et al., 2024). Foreground 3D representations $\mathbf{G}_f$ are reconstructed with Trellis (Xiang et al., 2025), and the background 3D representation $\mathbf{G}_b$ is produced using Viewcrafter (Yu et al., 2024).

**Baselines.** We compare CP4D against three categories of baselines: physics-driven simulation methods, conditional video generation models, and text-to-4D approaches. For physics-driven methods, we include PhysGen (Liu et al., 2024b), PhysGen3D (Chen et al., 2025a), and Omni-PhysGS (Lin et al., 2025). For conditional video generation, we evaluate open-source models such as CogVideoX (Yang et al., 2025) and Wan (Wan et al., 2025), as well as proprietary systems including Sora (OpenAI, 2024) and Runway (Runway, 2024). Finally, DreamGaussian4D (Ren et al., 2023) is selected as a representative text-to-4D baseline.

**Metrics.** To assess the quality of generated videos, we adopt VBench (Huang et al., 2024) for evaluating motion smoothness, subject consistency, and image quality. In addition, WorldScore (Duan et al., 2025) is employed to measure photo consistency, 3D consistency, and motion smoothness. To further assess prompt adherence, following PhysGen3D (Chen et al., 2025a), we leverage GPT-4o to score generated videos across three dimensions: physical realism, photorealism, and semantic alignment with the input prompt. Please refer to more details in Appendix A.

### 5.2 COMPARISONS WITH STATE-OF-THE-ART METHODS

As illustrated in Fig. 4, we present two challenging cases for qualitative comparison. In the deformable object motion scenario (*i.e.*, the left side of Fig. 4), Sora (OpenAI, 2024) demonstrates limited capability in accurately identifying the target object and modeling its physical dynamics, and further synthesizes spurious motion patterns involving entities absent from the input image. PhysGen3D (Chen et al., 2025a) reconstructs 3D meshes with low geometric fidelity and spatial arrangements that violate physical plausibility, substantially degrading visual realism. Wan (Wan et al., 2025) exhibits pronounced temporal instability due to color flickering, and fails to respond to

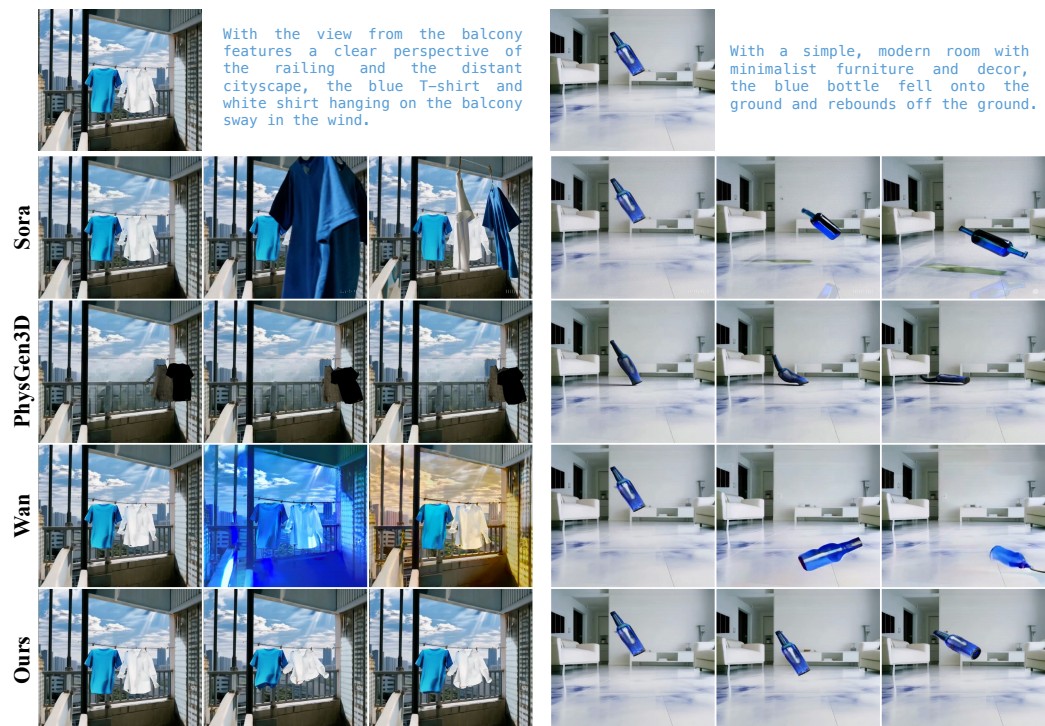

Figure 4: **Qualitative comparisons.** The top row shows the given text prompt and the corresponding generated image. Our method generates temporally consistent and physically plausible videos, outperforming baseline approaches in both visual fidelity and physical realism.

| Model | VBench | | | WorldScore | | |
|---|---|---|---|---|---|---|
| | Motion↑ | Consistency↑ | Imaging↑ | Photo Consist↑ | 3D Consist↑ | Motion Smooth↑ |
| Runway Runway (2024) | 0.995 | 0.936 | **0.644** | 62.66 | 86.34 | 68.43 |
| Sora (OpenAI, 2024) | 0.993 | 0.904 | 0.592 | 52.95 | 64.26 | 33.44 |
| CogVideoX-I2V-5B (Yang et al., 2025) | 0.993 | 0.932 | 0.603 | 70.06 | 81.90 | 73.66 |
| Wan2.2-TI2V-5B (Wan et al., 2025) | 0.991 | 0.934 | 0.599 | 72.66 | 77.50 | 47.04 |
| PhysGen (Liu et al., 2024b) | 0.996 | 0.966 | 0.621 | 88.34 | 90.04 | 81.67 |
| PhysGen3D (Chen et al., 2025a) | 0.997 | 0.963 | 0.599 | 93.07 | 92.99 | 90.95 |
| OmniPhysGS (Lin et al., 2025) | 0.995 | 0.960 | 0.356 | 22.54 | 48.80 | 92.88 |
| DreamGaussian4D (Ren et al., 2023) | 0.969 | 0.846 | 0.477 | 14.59 | 40.29 | 34.73 |
| **Ours** | **0.998** | **0.972** | 0.641 | **97.42** | **95.55** | **93.52** |

Table 1: **Quantitative comparisons. Bold**: Best. Underline: Second Best. Our proposed method consistently outperforms previous solutions on both VBench and WorldScore.

the motion prompt, resulting in static garments throughout the sequence. In contrast, our method produces coherent, artifact-free motion grounded in the input image, with significantly improved physical fidelity and temporal consistency. In the rigid-body collision scenario (*i.e.*, the right side of Fig. 4), PhysGen3D is restricted to elastic material simulation, causing the bottle to collapse unrealistically upon impact. Sora and Wan (Wan et al., 2025) further undermine plausibility by replacing the bottle with a different object post-collision, thereby breaking object identity and disrupting motion continuity. Compared to these methods, our approach preserves object identity throughout the interaction and yields physically consistent collision outcomes. Kindly refer to more results in Appendix E and F.

Quantitatively, as shown in Tab. 1, our method achieves superior motion coherence and temporal smoothness, consistently outperforming both video generative models and physics-driven methods across key dynamic metrics. Moreover, the generated videos exhibit high static visual quality, rivaling or even surpassing the strong closed-source baselines, particularly in terms of 3D consistency.

| Model | Physical realism↑ | Photorealism↑ | Semantic alignment↑ |
|---|---|---|---|
| Sora (OpenAI, 2024) | 0.547 | 0.729 | 0.665 |
| Runway Runway (2024) | 0.670 | 0.753 | 0.732 |
| Wan2.2-TI2V-5B (Wan et al., 2025) | 0.576 | 0.626 | 0.635 |
| PhysGen (Liu et al., 2024b) | 0.524 | 0.615 | 0.588 |
| PhysGen3D (Chen et al., 2025a) | 0.624 | 0.624 | 0.626 |
| OmniPhysGS (Lin et al., 2025) | 0.347 | 0.265 | 0.170 |
| DreamGaussian4D (Ren et al., 2023) | 0.229 | 0.112 | 0.176 |
| **Ours** | **0.694** | **0.759** | **0.747** |

Table 2: **GPT-4o Evaluation Results**. Our proposed method can achieve the best results.

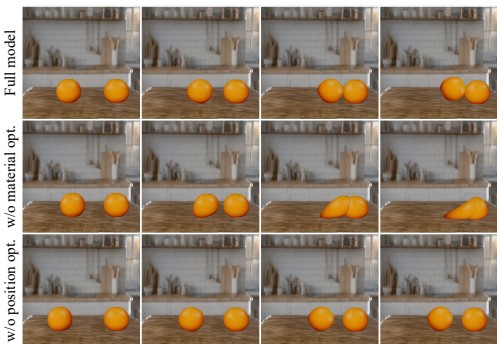

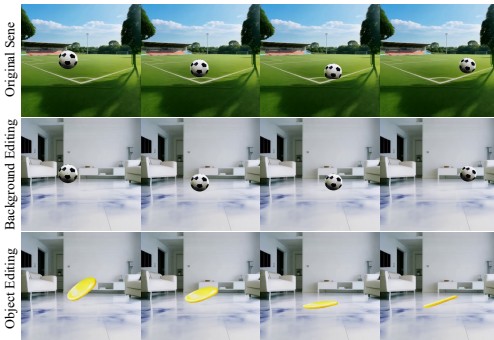

Figure 5: **Ablation study.** Results of ablation on optimizing VLM-estimated physical parameters and foreground object positions.

Figure 6: **Editing results.** Examples of background environment and foreground object editing in generated 4D scenes.

Regarding physical plausibility, as demonstrated in Tab. 2, our method surpasses all competing approaches on the physics realism metric, while simultaneously maintaining strong alignment with the input text, thereby ensuring high semantic consistency.

### 5.3 ABLATION STUDY

As illustrated in Sec. 4.2, to address inaccuracies in VLM-estimated physical parameters and the limited precision of physics simulators, we employ SDS to separately optimize the material parameters predicted by VLMs and the relative positions of foreground objects. To verify the necessity of these designs, we provide ablation studies here. As shown in Fig. 5, omitting material optimization causes the VLM-predicted density and Young's modulus to yield overly compliant simulations, leading to unstable or non-physical object motion. Without relative position optimization, the simulation of multi-object interactions produces spurious collisions in the absence of true spatial overlap. When both optimization modules are applied, our method yields more stable dynamics and visually plausible object interactions. More ablation studies are provided in the Appendix D.

### 5.4 APPLICATIONS ON CONTROLLABLE EDITING

The compositional design of our method endows it with the inherent ability to edit individual concepts, *e.g.*, varying background environments and foreground objects with distinct motions. As shown in Fig. 6, we can seamlessly replace them in a zero-shot manner while preserving scene coherence, physical plausibility, and temporal consistency, thereby enabling flexible and diverse 4D content generation.

## 6 CONCLUSION

In this work, we have presented CP4D, a novel framework for photorealistic 4D scene generation with faithful modeling of complex physical dynamics. Drawing inspiration from the compositional nature of real-world scenes, CP4D follows a three-stage pipeline: 1) constructing 3D representations of background environments and foreground objects from textual prompts using pre-trained expert models; 2) producing physically grounded trajectories and realistic interactions through a hybrid motion synthesis strategy; and (3) seamlessly integrating static environments with dynamic objects via an automated composition mechanism. Extensive experiments have demonstrated that our proposed method consistently outperforms state-of-the-art baselines.

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

# APPENDIX FOR CP4D: COMPOSITIONAL PHYSICS-AWARE 4D SCENE GENERATION

> With the view from the balcony features a clear perspective of the railing and the distant cityscape, the blue T-shirt and white shirt hanging on the balcony sway in the wind.

> With a simple, modern room with minimalist furniture and decor, the blue bottle fell onto the ground and rebounds off the ground.

Figure 7: Fixed structural tokens appear in black, background descriptors in red, and motion-related descriptors of foreground objects in blue.

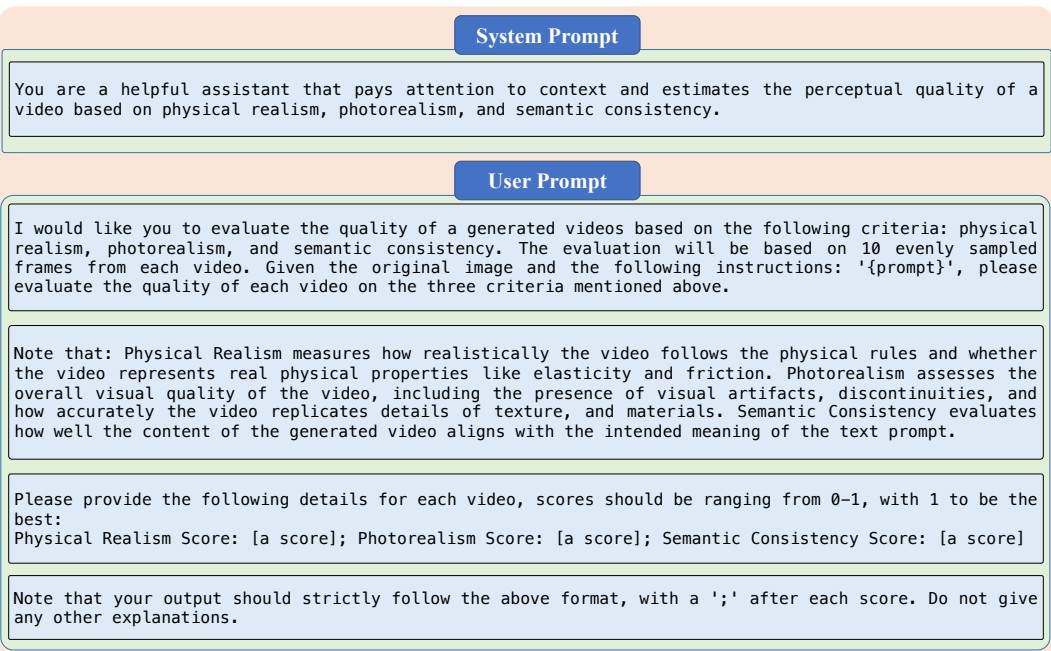

**System Prompt**

You are a helpful assistant that pays attention to context and estimates the perceptual quality of a video based on physical realism, photorealism, and semantic consistency.

**User Prompt**

I would like you to evaluate the quality of a generated videos based on the following criteria: physical realism, photorealism, and semantic consistency. The evaluation will be based on 10 evenly sampled frames from each video. Given the original image and the following instructions: '{prompt}', please evaluate the quality of each video on the three criteria mentioned above.

Note that: Physical Realism measures how realistically the video follows the physical rules and whether the video represents real physical properties like elasticity and friction. Photorealism assesses the overall visual quality of the video, including the presence of visual artifacts, discontinuities, and how accurately the video replicates details of texture, and materials. Semantic Consistency evaluates how well the content of the generated video aligns with the intended meaning of the text prompt.

Please provide the following details for each video, scores should be ranging from 0-1, with 1 to be the best:
Physical Realism Score: [a score]; Photorealism Score: [a score]; Semantic Consistency Score: [a score]

Note that your output should strictly follow the above format, with a ';' after each score. Do not give any other explanations.

Figure 8: Prompt used for GPT-4o evaluation.

## A MORE EXPERIMENTAL DETAILS

### A.1 DATASET CURATION

To construct a dataset capable of comprehensively evaluating adherence to physical laws, we adapt textual prompts from VideoPhy Bansal et al. (2024) and design a set of 17 representative prompts. These prompts ensure coverage of at least two distinct examples for each physical category, including rigid-body, elastic, deformable, and fluid dynamics. All prompts follow a standardized format that explicitly defines both the background context and the motion of foreground objects, as illustrated in Fig. 7. Given that several baseline methods operate in image-to-video or 3D-to-video paradigms, we establish unified image and 3D foreground representations to enable fair comparison. We employ the Qwen-Image Wu et al. (2025) model to generate initial images conditioned on each text prompt. We then apply Qwen-Image-Edit to produce corresponding background-only images by removing foreground content. To obtain 3D representations of foreground objects, we first apply the SAM Kirillov et al. (2023) model to segment candidate regions and combine it with a VLM to automatically select the segmentation mask that best aligns with the input text. This process

yields a cropped image containing only the target foreground object. We subsequently use Trellis to reconstruct a 3D-GS representation, denoted as $\mathbf{G}_f$, from the segmented foreground image. For OmniPhysGS Lin et al. (2025), which supports only 3D-to-video generation, we use the resulting $\mathbf{G}_f$ as input for physics simulation. Since this method does not support background conditioning, its output is restricted to motion sequences rendered against a blank background. Similarly, Dream-Gaussian4D Ren et al. (2023) inherently isolates the foreground during generation and produces only 4D motion outputs of the foreground object without incorporating background context.

### A.2 Implementation details

All experiments are conducted on NVIDIA A800 GPUs, each equipped with 40 GB of memory. For material optimization, we employ distributed training across 8 GPUs to accelerate convergence, whereas position optimization and all inference tasks are performed on a single GPU for efficiency. We train the model using the Adam optimizer Kingma (2014), with 5 epochs dedicated to material optimization and 100 epochs for position optimization. For the initialization of material properties and external force conditions, our framework offers flexible support for both automatic estimation using VLMs and manual specification by the user. For instance, in simulating elastic collisions, users may freely select constitutive models (*e.g.*, elastic or plastic) and manually specify physical parameters such as density, Young's modulus, and Poisson's ratio. These user-provided material attributes are subsequently refined during training to ensure the rendered output aligns perceptually with visual expectations. To ensure spatial alignment with the input image, we render the full scene from the camera viewpoint corresponding to the initial configuration of the background model.

### A.3 Evaluation by GPT-4o

Since no widely accepted, physics-focused evaluation metrics currently exist, we follow the approach of PhysGen3D Chen et al. (2025a) and additionally employ GPT-4o to conduct subjective assessments. These evaluations score multiple dimensions relevant to physical plausibility, including physical realism, photorealism, and semantic alignment. We adopt the evaluation prompt design originally proposed in PhysGen3D Chen et al. (2025a); as their work has empirically demonstrated strong alignment with human judgment, we apply only minor adaptations to tailor the prompt to our dataset, without introducing substantial modifications. The complete evaluation prompt is illustrated in Fig. 8.

## B Automated initialization estimation of materials and external forces

To enable more intelligent initialization of an object's material properties and external force conditions aligned with task requirements, we leverage GPT-4o to estimate material parameters—particularly for elastic and deformable materials, for which appropriate elastic and plastic constitutive models must be selected. We have designed a dedicated prompt template for material estimation; an example prompt targeting elastic materials is illustrated in Fig.9. Regarding external loading conditions, we estimate initial parameters including gravity, additional external forces, initial velocities, and damping coefficients. The corresponding prompt template for these conditions is presented in Fig.10.

## C Heterogeneous physical-solvers

### C.1 MPM solver for elastic and flexible objects

To simulate the dynamic behavior of elastic and flexible materials, we employ the Material Point Method (MPM), a hybrid computational framework that combines the strengths of Lagrangian particle descriptions and Eulerian grid-based solvers. The continuum is discretized into a collection of particles, each representing a small material region. These particles track several time-varying Lagrangian quantities, such as position $x_p$, velocity $v_p$, and deformation gradient $F_p$. The conservation of mass within the Lagrangian particles ensures the invariance of total mass during movement. In contrast, the conservation of momentum is more intuitively represented in an Eulerian framework,

```
You are a materials science expert. Based on the object name and description below, estimate its
physical parameters and select appropriate elasticity/plasticity model combinations for simulation.

Object Name: {object_name}
Object Description: {object_description}

Return exactly one line in this strict format:
density,Young's_modulus,Poisson_ratio,"[elasticity_model1,elasticity_model2,...]","[plasticity_model1,p
lasticity_model2,...]"

Parameter Ranges:
● Density: 50–20000 kg/m³
● Young's Modulus: 1e5–1e12 Pa
● Poisson's Ratio: 0.0–0.5
Model Options:
● Elasticity: SigmaElasticity, CorotatedElasticity, StVKElasticity, VolumeElasticity, FluidElasticity
● Plasticity: IdentityPlasticity, DruckerPragerPlasticity, SigmaPlasticity, VonMisesPlasticity

Examples:
● Rubber: 1200,1000000,0.49,"[SigmaElasticity,CorotatedElasticity]","[SigmaPlasticity]"
● Water: 1000,2000000000,0.5,"[FluidElasticity,VolumeElasticity]","[IdentityPlasticity]"

Note: Do not explain. Do not add units. Do not use line breaks. Output only the comma-separated line.
```

Figure 9: Prompt template for automated material estimation.

```
You are a physics simulation expert specializing in intelligent physical parameter configuration for
physical simulation. Given the original image, the following instruction '{prompt}', the object's
spatial extent and the definition of coordinate system:
● +X axis: Horizontal direction pointing leftward in the image plane.
● +Y axis: Vertical direction pointing upward in the image plane.
● +Z axis: Orthogonal to the image plane (depth direction).

Please finish the following tasks:

Task:
Based on the object's geometric properties and the intended simulation objective, determine the
following physical parameters:
● Gravity: According to image and instruction, is gravity required? (Yes/No). If yes, specify its 3D
  vector direction (e.g., [0, −9.8, 0] for downward gravity) and magnitude (in m/s²).
● External Force: Whether to apply forces other than gravity to the object; if needed, please set the
  range and magnitude of the force according to the object's spatial extent.
● Initial Velocity: Is an initial velocity required? (Yes/No). If yes, specify its 3D vector direction
  (e.g., [2.0, 0, 0] for rightward motion) and magnitude (in m/s).
● Simulation Parameters: Recommend key parameters such as damping coefficients (e.g., global damping,
  friction), time step size, justifying your choices based on stability and physical realism.

Please return the configuration in JSON format:
{
    "gravity": [x, y, z],
    "force 1": {"force": [x, y, z], "region": {[x1, x2], [y1, y2], [x3, y3]}} ,
    "initial_velocity": [x, y, z],
    "simulation_params": {"damping": 0.8, "dt": 0.001}
}
```

Figure 10: Prompt template for automated external forces estimation.

which circumvents the need for mesh construction. We adhere to the methodology of Stomakhin et al. Stomakhin et al. (2013) to integrate these representations using $C^1$ continuous B-spline kernels for bidirectional transfer. From time step $t_n$ to $t_{n+1}$, the momentum conservation, discretized via the forward Euler scheme, is expressed as:

$$m_i \frac{(v_i^{n+1} - v_i^n)}{\Delta t} = -\sum_p V_p^0 \frac{\partial \Psi}{\partial F}(F_{pE,n}) F_{pE,n}^T \nabla w_{ip}^n + f_i^{\text{ext}}, \tag{10}$$

here, $i$ and $p$ denote the fields on the Eulerian grid and the Lagrangian particles, respectively; $w_{ip}^n$ is the B-spline kernel defined on the $i$-th grid evaluated at $x_p^n$; $V_p^0$ represents the initial volume, and $\Delta t$ is the time step size. The updated grid velocity field $v_i^{n+1}$ is subsequently transferred back to the particle velocity $v_p^{n+1}$, updating the particle positions to $x_p^{n+1} = x_p^n + \Delta t v_p^{n+1}$. We track $F_E$ rather than both $F$ and $F_P$ Simo & Hughes (2006), with updates to $F_{E,p}^{n+1}$ achieved via:

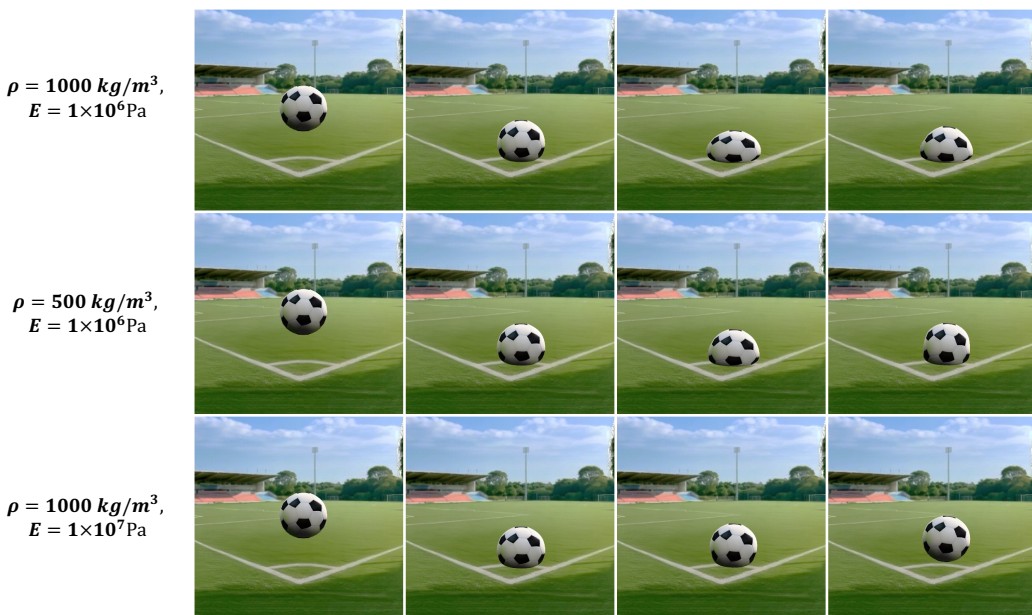

Figure 11: Physical simulation results of elastic bodies under different density and Young's modulus.

$$F_{E,p}^{n+1} = (I + \Delta t \nabla v_p) F_{E,p}^n = \left( I + \Delta t \sum_i v_i^{n+1} \nabla w_{ip}^{n,T} \right) F_{E,p}^n. \tag{11}$$

This is regularized through an additional return mapping to facilitate plasticity evolution: $F_{E,p}^{n+1} \leftarrow \mathcal{Z}(F_{E,p}^{n+1})$.

While the aforementioned methods effectively simulate the dynamics of individual objects, they exhibit limitations when handling multi-object interactions. In the MPM solver, spatial discretization via Eulerian grids inherently introduces slight geometric expansion of material regions. As a result, during simulation rendering, collisions may visually manifest as "phantom contacts," where objects appear to interact prior to actual physical contact, as illustrated in Fig. 2. To mitigate this positional artifact in multi-object scenarios, we introduce a post-simulation position refinement module specifically designed for MPM outputs. This module optimizes the visual trajectories using an SDS loss, thereby enhancing spatial accuracy and perceptual realism in the rendered results.

One critical factor limiting the fidelity of MPM simulations is the inappropriate specification of material properties. For instance, when an object is assigned a high density but an excessively low Young's modulus, it exhibits unrealistically large deformations and overly soft behavior, as illustrated in the first row of Fig. 11. Through optimization of the physical parameters, particularly by increasing the Young's modulus to better reflect material stiffness, we obtain simulation results that are significantly more realistic and physically plausible.

### C.2 RIGID COLLISION SIMULATION

Our proposed method integrates traditional rigid body dynamics with 3D Gaussian particle rendering, establishing a unified framework for physics simulation and visual rendering. The core innovation lies in representing complex 3D objects as collections of Gaussian particles while maintaining rigid body constraints for physical accuracy.

**1) Basic motion representation**

The rigid body dynamics follow the Newton-Euler equations: Translational Motion:

$$\mathbf{F} = m\dot{\mathbf{v}}, \dot{\mathbf{x}} = \mathbf{v}, \tag{12}$$

Figure 12: Multi-puck rigid-body collision cases in shuffleboard simulation.

Rotational Motion:

$$\boldsymbol{\tau} = I\dot{\boldsymbol{\omega}}, \dot{\mathbf{q}} = \frac{1}{2}\mathbf{q} \otimes \boldsymbol{\omega}, \tag{13}$$

where $\otimes$ denotes quaternion multiplication, and $\mathbf{F}$ and $\boldsymbol{\tau}$ represent the net force and torque acting on the rigid body, respectively.

A key contribution of our approach is the computation of the inertia tensor based on the actual Gaussian particle distribution rather than simplified geometric approximations:

$$I = \sum_i m_i(r_i^2\mathbf{I} - \mathbf{r}_i \otimes \mathbf{r}_i), \tag{14}$$

where $m_i$ is the mass of the $i$-th Gaussian particle, $\mathbf{r}_i$ is the position vector relative to the center of mass, and $\mathbf{I}$ is the identity matrix. This formulation ensures that the rotational dynamics accurately reflect the object's true geometric distribution.

**2) Collision detection and response**

**Ground Collision Detection.** Our system implements precise collision detection based on the Gaussian particle distribution. For ground collisions, the system identifies contact particles:

$$\mathcal{C} = \{\mathbf{x}_i : y_i \le h_{ground} + \epsilon\}, \tag{15}$$

where $h_{ground}$ is the ground height and $\epsilon$ is a small tolerance for numerical stability.

**Impulse-Based Collision Response.** The collision response employs an impulse-based approach: Linear Impulse:

$$J = -(1+e)\frac{\mathbf{v} \cdot \mathbf{n}}{1/m_1 + 1/m_2}, \tag{16}$$

Angular Impulse:

$$\boldsymbol{\tau} = \mathbf{r} \times \mathbf{J}, \tag{17}$$

where $\mathbf{r}$ is the vector from the center of mass to the collision point, and $e$ is the coefficient of restitution.

**Inter-Body Collision Detection.** The system implements pairwise collision detection between rigid bodies:

$$d_{ij} = \|\mathbf{x}_i - \mathbf{x}_j\| - (r_i + r_j), \tag{18}$$

where $d_{ij}$ is the separation distance between bodies $i$ and $j$, and $r_i, r_j$ are their respective radii.

### C.3 POSITION-BASED DYNAMICS SOLVER FOR FLUID OBJECTS

The PBD framework operates on the fundamental principle of constraint satisfaction rather than force integration. For fluid simulation, the primary constraint is density preservation:

$$C_i(\mathbf{x}_1, \mathbf{x}_2, ..., \mathbf{x}_n) = \rho_i - \rho_0, \tag{19}$$

where $\rho_i$ is the current density of particle $i$, and $\rho_0$ is the target density. The position correction is computed as:

$$\Delta\mathbf{x}_i = \frac{1}{\lambda_i}\nabla C_i, \tag{20}$$

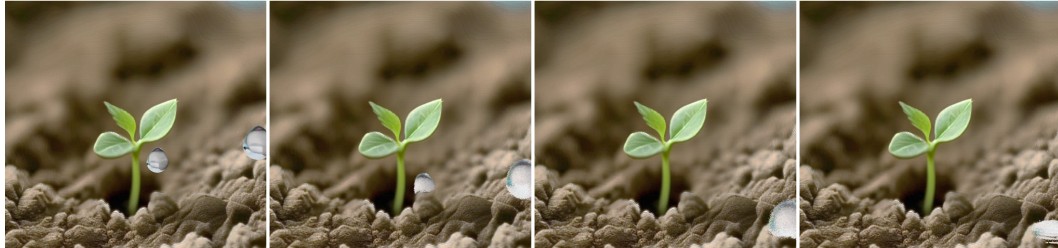

Figure 13: An example of raindrop falling simulated using the PBD solver. Raindrops are visually scaled up here to facilitate observation of fluid motion dynamics.

where $\lambda_i$ is the Lagrange multiplier associated with the density constraint.

**1) Our PBD solver**

Our PBD fluid solver incorporates multiple physical effects through configurable parameters:

Cohesion Force:

$$\mathbf{F}_{cohesion} = \alpha_{cohesion} \cdot (\mathbf{x}_{center} - \mathbf{x}_i) \cdot \|\mathbf{x}_{center} - \mathbf{x}_i\|, \tag{21}$$

where $\alpha_{cohesion}$ is the cohesion strength parameter, and $\mathbf{x}_{center}$ is the center of mass of all particles.

Surface Tension:

$$\mathbf{F}_{surface} = \alpha_{surface} \cdot (\mathbf{x}_{local} - \mathbf{x}_i), \tag{22}$$

where $\mathbf{x}_{local}$ is the local center of neighboring particles within a radius $r$, and $\alpha_{surface}$ controls surface tension strength.

Turbulence Force:

$$\mathbf{F}_{turbulence} = \alpha_{turbulence} \cdot \begin{bmatrix} \sin(\omega_y y_i) \\ \cos(\omega_z z_i) \\ \sin(\omega_x x_i) \end{bmatrix}, \tag{23}$$

where $\omega_x, \omega_y, \omega_z$ are spatial frequencies, and $\alpha_{turbulence}$ controls turbulence intensity.

**2) Efficiency optimization.** For computational efficiency, we employ a simplified neighbor search strategy:

$$\mathcal{N}_i = \{j : \|\mathbf{x}_i - \mathbf{x}_j\| < r_{search}\}, \tag{24}$$

where $r_{search}$ is the search radius, typically set to $3 \cdot r_{particle}$. The neighbor weights are computed as:

$$w_{ij} = \frac{1}{1 + \|\mathbf{x}_i - \mathbf{x}_j\|/r_{particle}}. \tag{25}$$

**3) Constraint Projection Algorithm Density Constraint Projection.** The density constraint projection follows an iterative approach: Lagrange Multiplier Computation:

$$\lambda_i = -\frac{C_i(\mathbf{x})}{\sum_j \|\nabla C_i\|^2 + \epsilon}. \tag{26}$$

Position Correction:

$$\Delta \mathbf{x}_i = \lambda_i \nabla C_i. \tag{27}$$

Iterative Refinement:

$$\mathbf{x}_i^{k+1} = \mathbf{x}_i^k + \Delta \mathbf{x}_i, \tag{28}$$

where $k$ denotes the iteration number, and $\epsilon$ is a small regularization term.

**Boundary Constraint Handling.** Boundary constraints are enforced through position clamping and velocity reflection: Position Clamping:

$$\mathbf{x}_i^{clamped} = \mathrm{clamp}(\mathbf{x}_i, \mathbf{x}_{min}, \mathbf{x}_{max}). \tag{29}$$

Velocity Reflection:

$$\mathbf{v}_i^{reflected} = \mathbf{v}_i \cdot (-\mu_{bounce}), \tag{30}$$

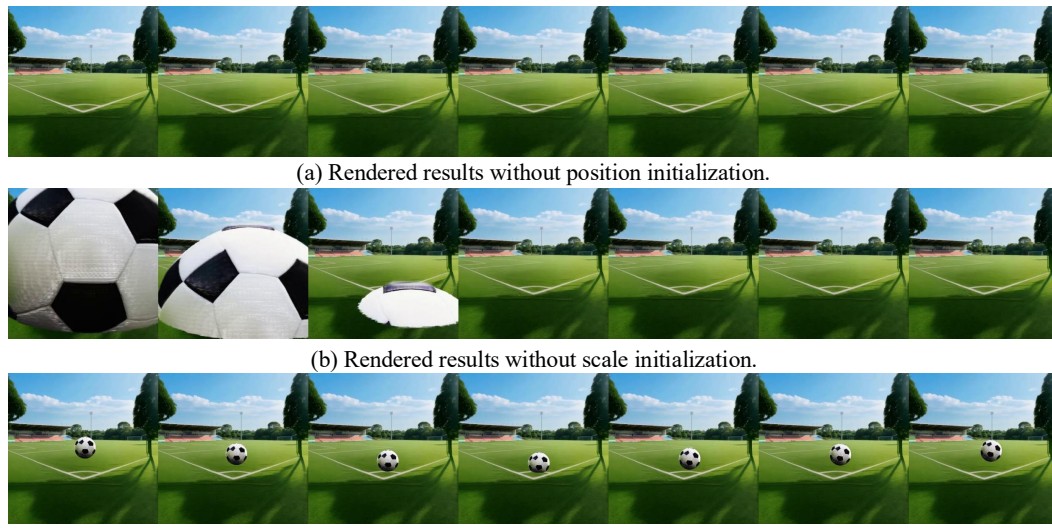

(a) Rendered results without position initialization.

(b) Rendered results without scale initialization.

(c) Rendered results with both position and scale initialization.

Figure 14: Qualitative comparisons of ablation studies on the position initialization and scale initialization illustrated in Sec. 4.3.

| Metrics | Refinement (Sec. 4.2) | | | Composition (Sec. 4.3) | | |
|---|---|---|---|---|---|---|
| | w/o $\mathcal{O}_m$ | w/o $\mathcal{O}_o$ | ours | w/o $\mathcal{O}_s$ | w/o $\mathcal{O}_p$ | ours |
| Motion | 0.955 | 0.957 | 0.958 | 0.509 | 0.939 | 0.975 |
| Consistency | 0.962 | 0.994 | 0.995 | 0.689 | 0.999 | 0.989 |
| Imaging | 0.556 | 0.627 | 0.628 | 0.549 | 0.543 | 0.640 |
| PhysReal | 0.5 | 0.0 | 0.8 | 0.4 | 0.0 | 0.7 |
| PhotoReal | 0.7 | 0.8 | 0.9 | 0.7 | 0.8 | 0.8 |
| Align | 0.6 | 0.0 | 1.0 | 0.6 | 0.0 | 0.9 |

Table 3: **Quantitative ablation study results**. Motion Smoothness (Motion) is evaluated using the WorldScore benchmark, while Subjective Consistency (Consistency) and Imaging quality (Imaging) metrics are derived from VBench. Physical Realism (PhysReal), Photographic Realism (PhotoReal), and Semantic Alignment (Align) are assessed by GPT-4o.

where $\mu_{bounce}$ is the boundary bounce coefficient.

**Diverse motion effects**

The solver incorporates directional flow effects:

$$\mathbf{F}_{flow} = \alpha_{flow} \cdot \mathbf{d}_{flow}$$

where $\mathbf{d}_{flow}$ is the flow direction vector, and $\alpha_{flow}$ controls flow strength.

Random perturbations are added to simulate natural fluid irregularities:

$$\mathbf{F}_{noise} = \alpha_{noise} \cdot \mathcal{N}(0, \mathbf{I})$$

where $\mathcal{N}(0, \mathbf{I})$ represents Gaussian noise with zero mean and unit variance.

## D  ADDITIONAL ABLATION STUDIES

To further validate the effectiveness of our design choices, we conduct additional ablation studies. As illustrated in Fig. 14, during the stage where background environments and foreground objects are composed into a coherent 4D scene (*i.e.*, Sec. 4.3), omitting position initialization from the monocular depth estimator causes foreground objects to be incorrectly occluded by the background,

thereby degenerating the output into a purely 3D scene. Similarly, removing scale initialization results in oversized foreground objects, which introduce severe visual artifacts.

As shown in Tab. 3, we also report quantitative comparisons. For the video generative model–based refinement described in Sec. 4.2, omitting the physical material refinement ($\mathcal{O}_m$) or the relative position refinement ($\mathcal{O}_o$) leads to clear performance degradation across multiple metrics, while our full model achieves the best results. Similarly, for the composition stage described in Sec. 4.3, removing the scale initialization ($\mathcal{O}_s$) or the position initialization ($\mathcal{O}_p$) ubstantially degrades performance. These results highlight the necessity of both refinement and composition strategies for producing physically plausible and visually coherent 4D scenes.

## E    MORE VISUAL COMPARISONS WITH BASELINES

As shown in Fig. 15, Fig. 16, Fig. 17, and Fig. 18, we provide additional visual comparisons with all baseline methods. These results highlight that our method achieves state-of-the-art quality, characterized by high visual fidelity, strong physical plausibility, and fine-grained controllability.

## F    RESULTS OF RENDERED MULTI-VIEW VIDEOS BY GENERATED 4D SCENE

As shown in Fig. 19, Fig. 20, Fig. 21, and Fig. 22, our method is able to generate complete 4D scenes and render consistent and physically plausible multi-view videos.

## G    LIMITATIONS AND FUTURE WORKS

Our approach adopts a stage-wise optimization strategy for each scene, which leads to relatively long runtimes when generating a complete physically realistic 4D scene. In future work, we plan to leverage the proposed framework to construct a large-scale 4D physical dataset and use it to train feed-forward generative models for 4D scene synthesis.

## THE USE OF LARGE LANGUAGE MODELS (LLMS)

In this paper, LLMs were only used to correct grammar and improve readability. They were not employed in any other part of the research process.

## ETHICS STATEMENT

This work does not involve human subjects, animal experiments, or the use of personally identifiable or sensitive information. The research presents no known risks of harm, bias, discrimination, or misuse, and raises no concerns regarding privacy, security, legal compliance, or research integrity. The authors have no conflicts of interest or external sponsorship that could influence the research outcomes.

## REPRODUCIBILITY STATEMENT

We have taken steps to ensure the reproducibility of our results. All experimental details and implementation choices are described in the main text and appendix. The complete source code will be made publicly available upon publication to facilitate reproduction and further research.

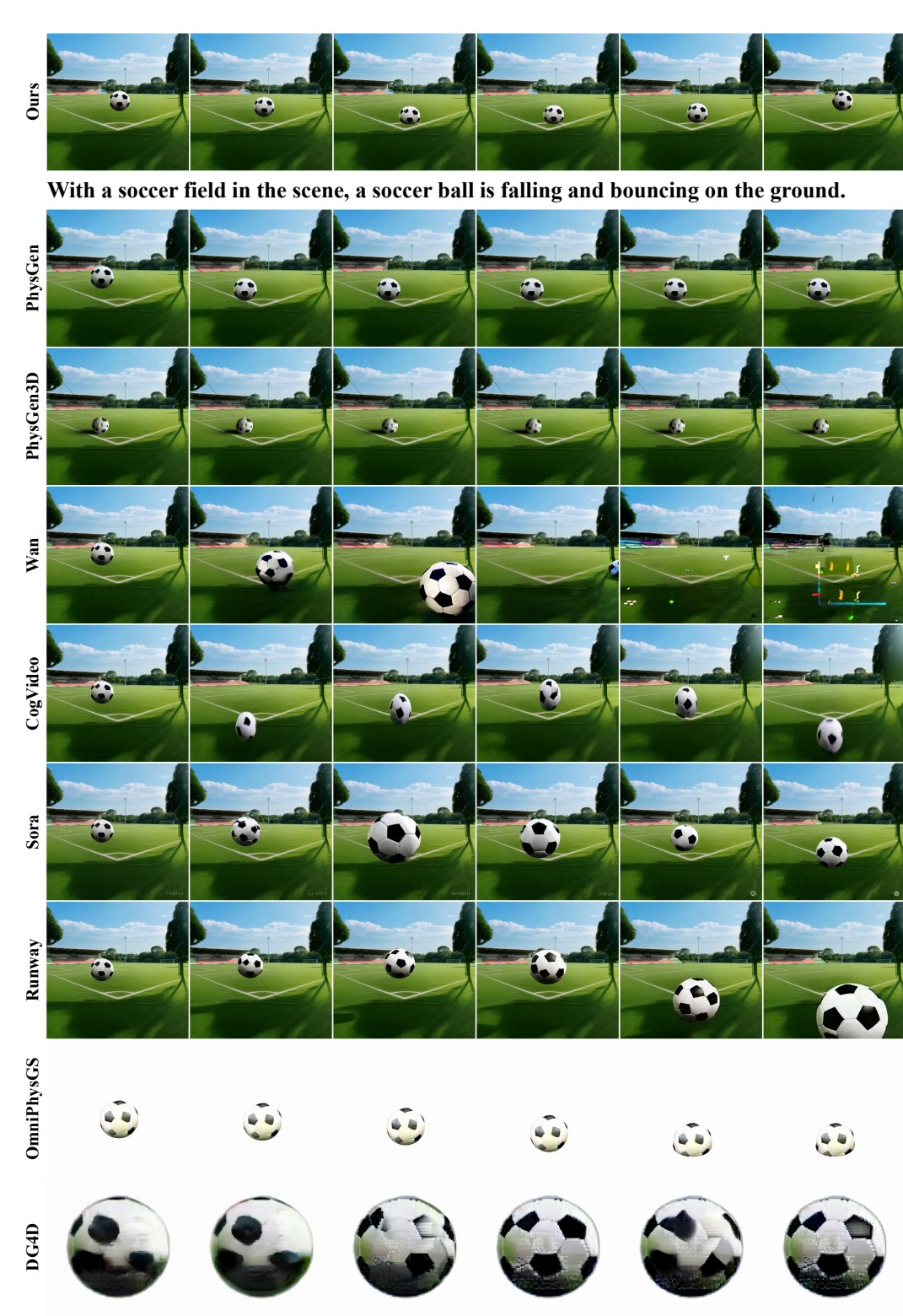

**With a soccer field in the scene, a soccer ball is falling and bouncing on the ground.**

Figure 15: Qualitative visualization results of our method and the baselines.

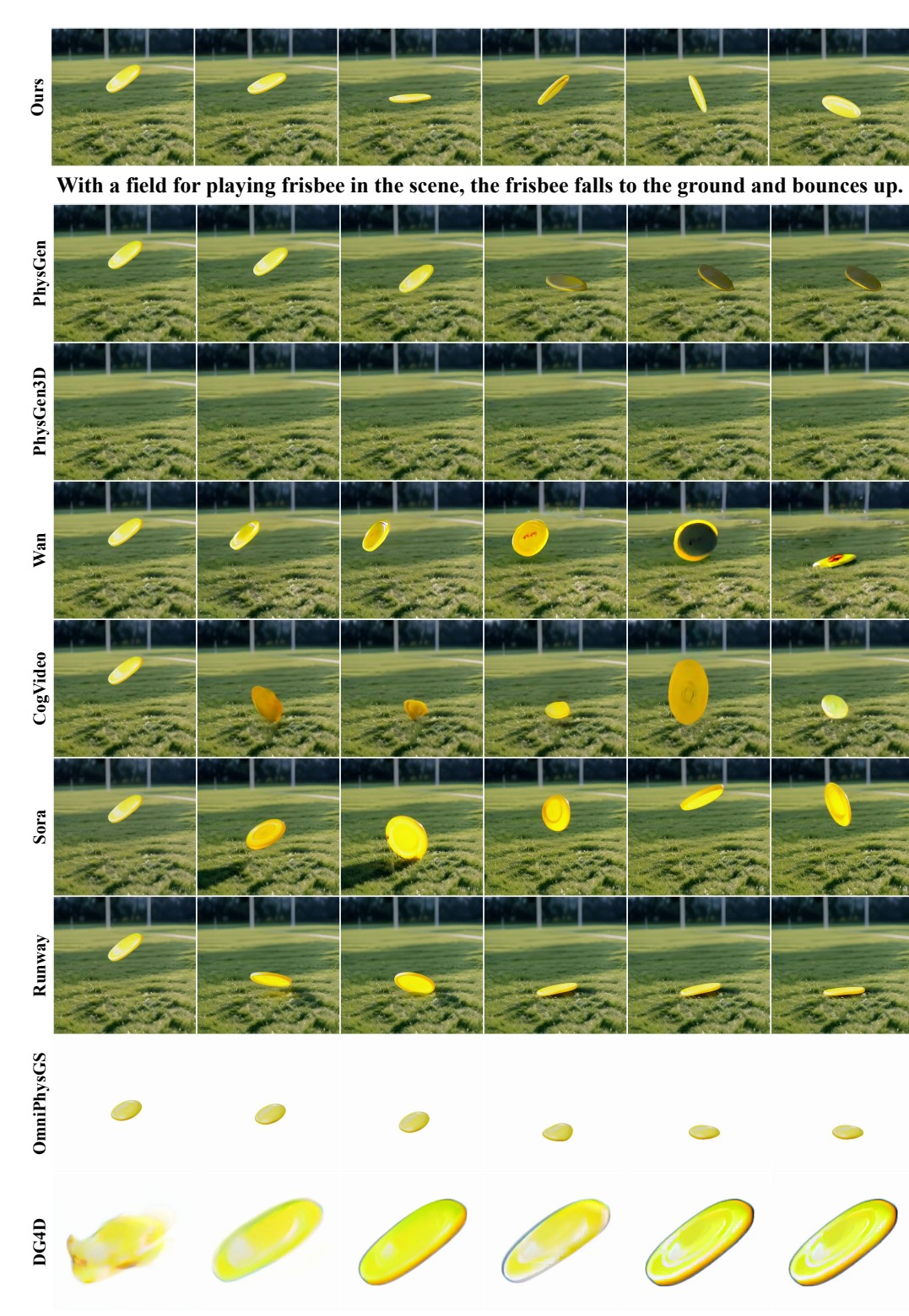

Figure 16: Qualitative visualization results of our method and the baselines.

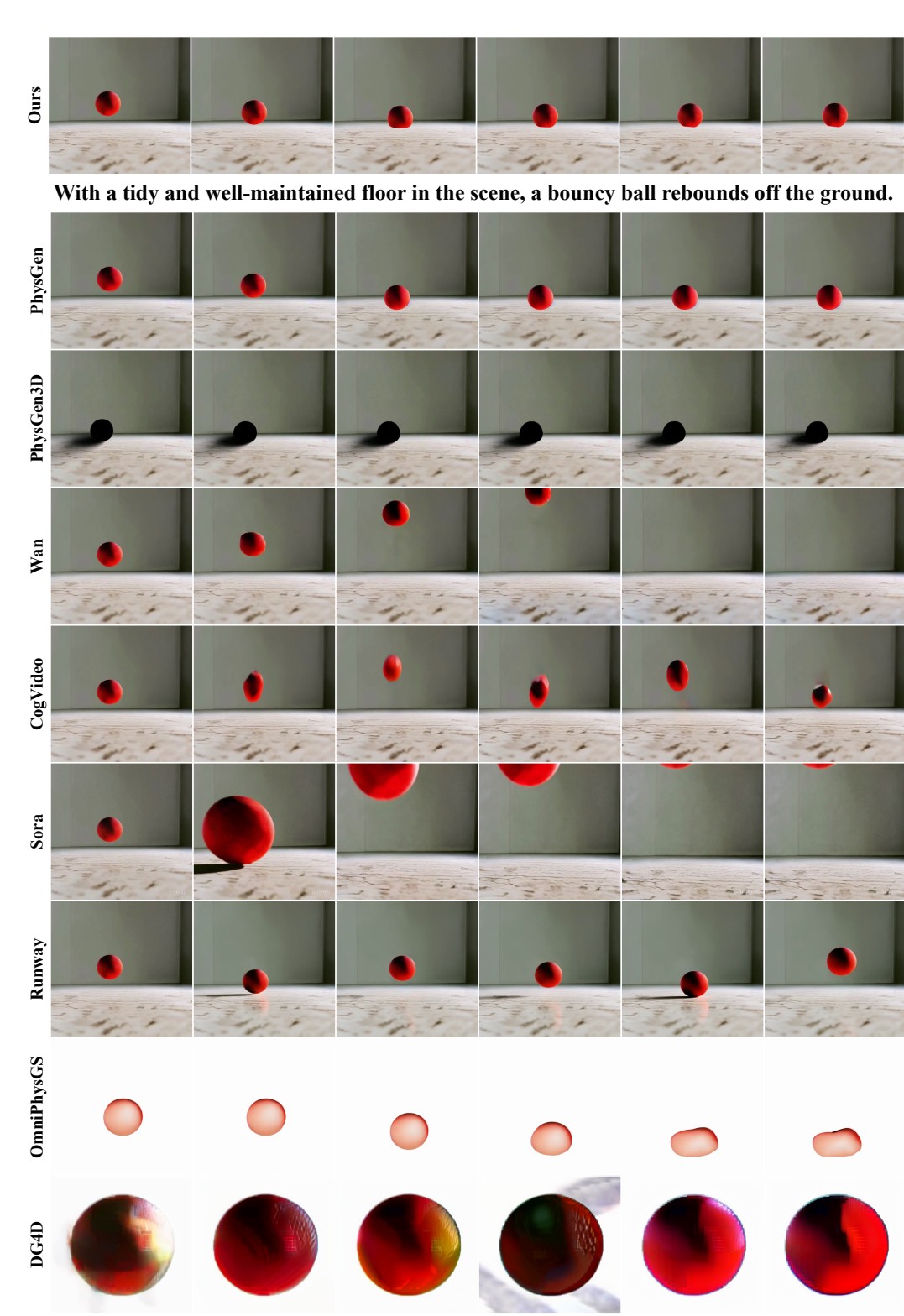

Figure 17: Qualitative visualization results of our method and the baselines.

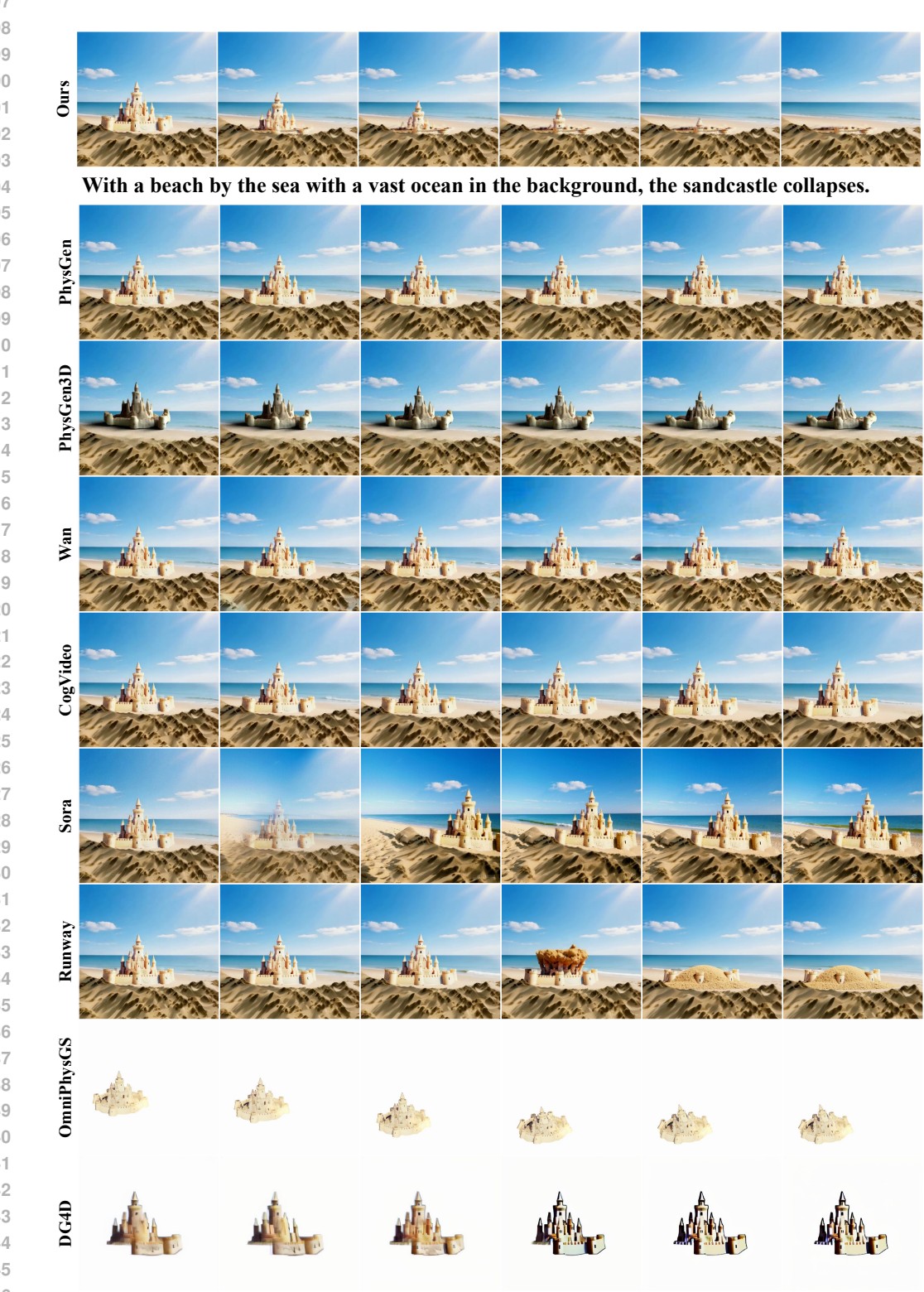

Figure 18: Qualitative visualization results of our method and the baselines.

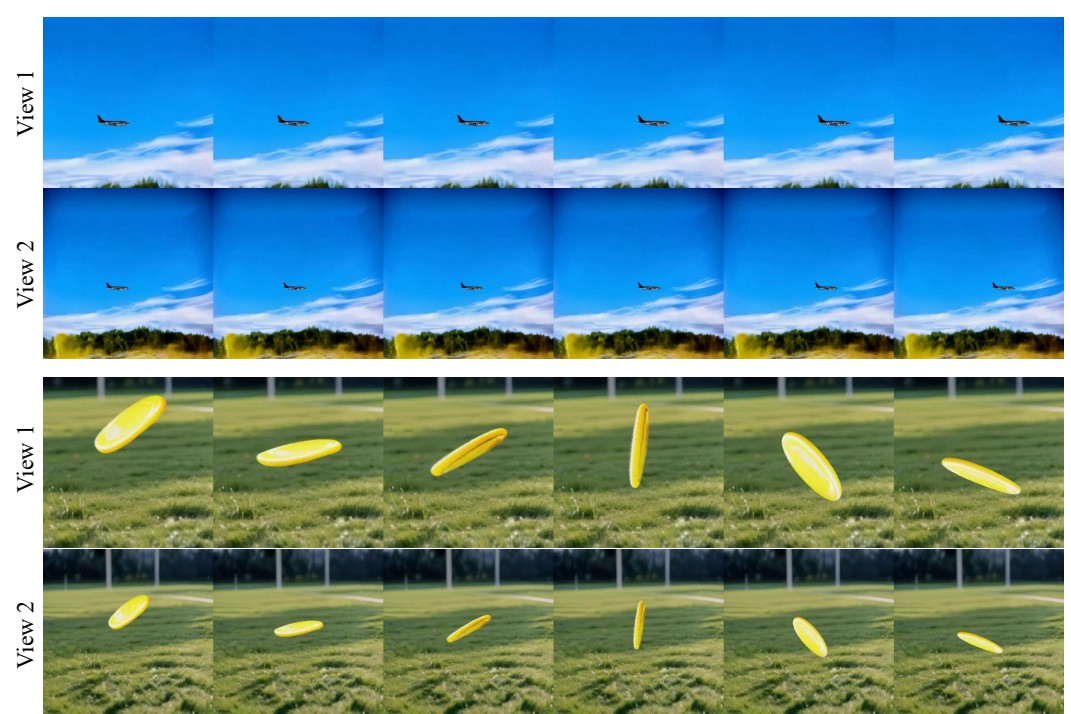

Figure 19: Visualization results of the motion process observed from different viewpoints

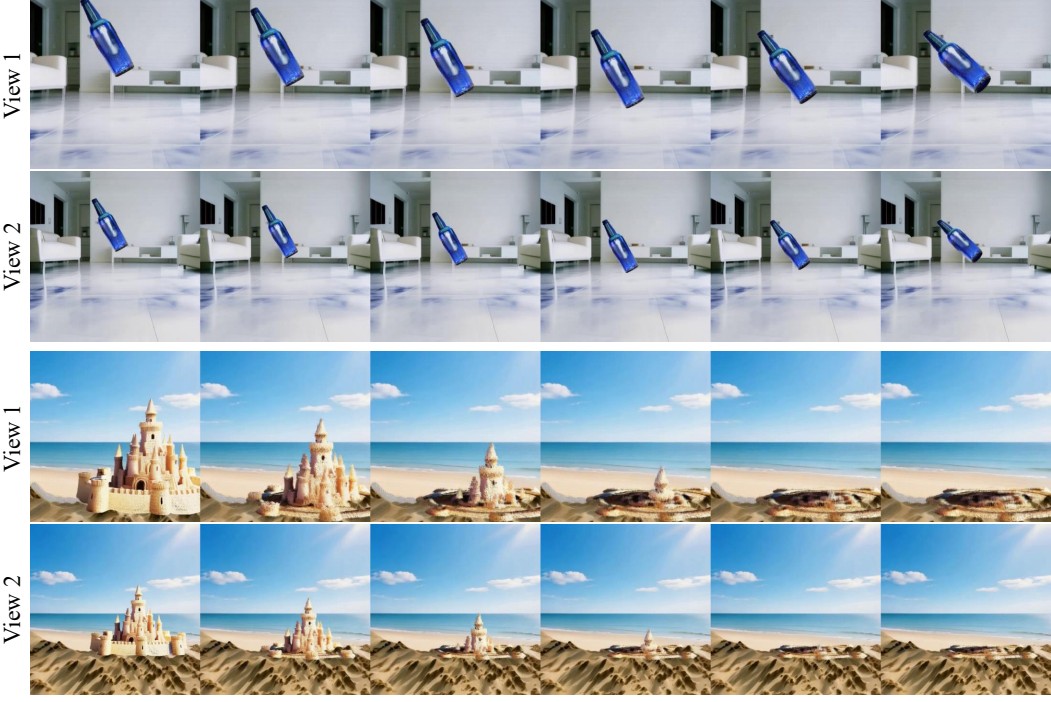

Figure 20: Visualization results of the motion process observed from different viewpoints

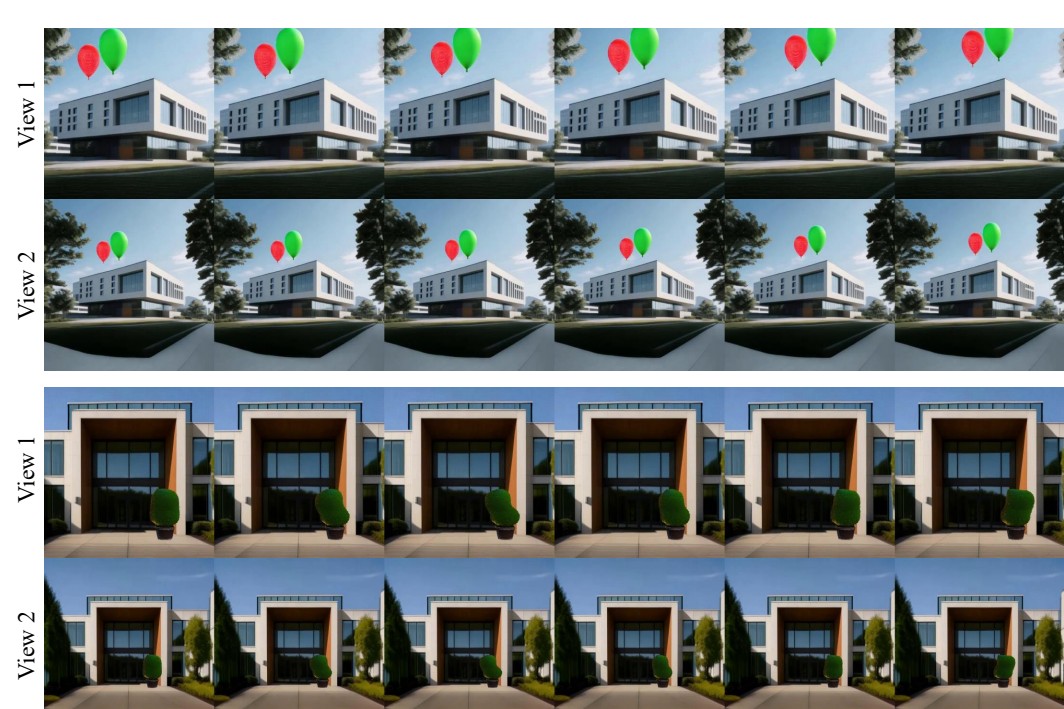

Figure 21: Visualization results of the motion process observed from different viewpoints

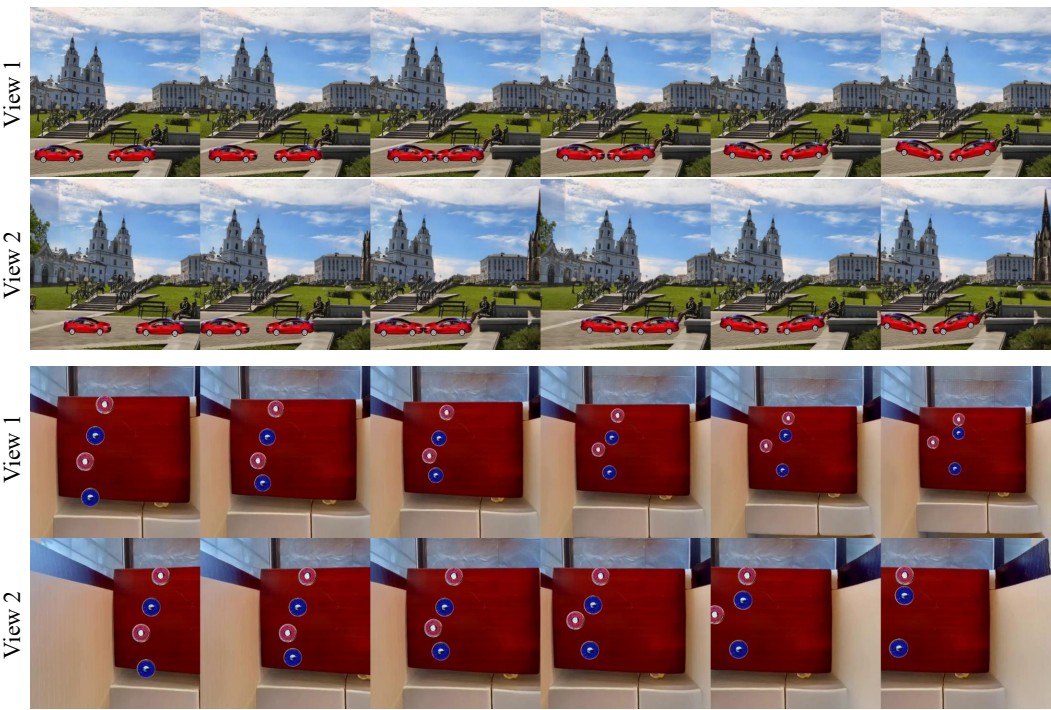

Figure 22: Visualization results of the motion process observed from different viewpoints

