# OpenReview forum: "CP4D: Compositional physics-aware 4D scene generation"
_ICLR.cc/2026/Conference — Submitted to ICLR 2026_

### Official Review · Reviewer_826b · 2025-10-27

**Soundness:** 2
**Presentation:** 3
**Contribution:** 1
**Rating:** 4
**Confidence:** 4

**Summary:**

The paper proposes CP4D, a compositional, physics-aware framework for 4D scene generation. The core idea is to decouple a static 3D background from physically grounded dynamic foreground objects, following a three-stage pipeline: (1) generate high-fidelity 3D representations for background and foreground with pre-trained expert models; (2) synthesize motion via heterogeneous physics simulators (MPM for elastic/flexible, rigid-body, and PBD for fluids) and then refine with video-diffusion SDS; (3) automatically compose foreground into background using monocular-depth–based position initialization and a camera-frustum–based scale heuristic, followed by optimization.

Experiments compare CP4D with physics-driven simulators, conditional video generators, and text-to-4D baselines, using VBench/WorldScore and an LLM-assisted evaluation; ablations indicate material and position optimization both contribute.

**Strengths:**

The pipeline of CP4D is clear and easy to follow. And for method, combining VLM-assisted physical initialization with SDS refinement is an interesting hybridization.

**Weaknesses:**

1. The motivation of CP4D is dividing static background and dynamic foreground, which is not novel in 4D generation area, such as [1,2,3].
2. The biggest concern of the pipeline is robustness. The pipeline depends on many off-the-shell models, especially the monocular depth  and a frustum heuristic part. The authors should give more examples to support the robustness of CP4D, while now only 17 simple prompts are listed.
3. The video results in supplementary material are not convincing, all cases just include very shot clip, and multiview examples do not show big camera motion exchange.
4. There is a typo in line75, where foreground is spelled as "foareground".

[1]. Comp4d: Llm-guided compositional 4d scene generation. Arxiv 24.03. Xu et al.
[2]. Compositional 3d-aware video generation with llm director. Nips2025. Zhu et al.
[3]. DynVideo-E: Harnessing Dynamic NeRF for Large-Scale Motion- and View-Change Human-Centric Video Editing. CVPR2024. Liu et al.

**Questions:**

Please see weakness.

---

> ### Author Response · Authors · 2025-11-23
>
> We sincerely appreciate the reviewer’s encouraging and valuable feedback. We are delighted that you find the overall pipeline clear and easy to follow, and we are encouraged by your recognition of our hybrid design that combines VLM-assisted physical initialization with SDS-based refinement. Your positive assessment is highly motivating for us. Below, we provide clarifications for the concerns raised.
>
> **Q1: Clarification on the novelty relative to prior works.**
>
> We sincerely thank the reviewer for bringing up these related works. As we explicitly state in the paper (Line 052), our formulation is indeed inspired by the compositional perspective explored in prior efforts such as Xu et al. (2024) and Zhu et al. (2024). However, these works address substantially different problem settings, whereas CP4D targets the far more challenging goal of text-driven, physics-aware 4D scene generation, which requires jointly ensuring physical plausibility, geometric consistency, and coherent foreground–background composition. Consequently, the novelty of CP4D does not lie in the high-level idea of separating static backgrounds from dynamic foregrounds, which we do not claim as new, but rather in how this decomposition is instantiated and extended to enable physically grounded 4D scene construction.
>
> Compared with these prior approaches, CP4D operates in a significantly broader and more physically grounded regime. For example, Comp4D relies almost entirely on VLMs to infer object attributes such as position, scale, and trajectory, an approach that introduces high uncertainty and limits applicability to relatively simple compositions. Other methods, such as Zhu et al. (2024), primarily focus on human-centric generation with SMPL-driven motion, where the foreground structure and motion space are predefined and constrained. In contrast, CP4D is designed to handle general objects across diverse materials, material-dependent physical behaviors, and complex scene geometries, enabling physically plausible dynamics and interactions that emerge from simulation rather than being directly predicted by a language model.
>
> In conclusion, CP4D provides a unified pipeline capable of producing physically coherent 4D scenes from a text prompt, a capability that goes well beyond what the cited compositional works are designed to achieve.
>
> **Q2: Clarification on the robustness of the CP4D pipeline.**
>
> We sincerely thank the reviewer for this important observation.  Despite involving multiple modules, CP4D is robust by design. In practice, the expert models we rely on, such as monocular depth estimators, segmentation models, and image/video/3D generative models, benefit from large-scale pretraining and exhibit strong empirical stability; we rarely observe failures from these components in our experiments. For the few stages that may introduce instability, such as VLM-based material parameter estimation, numerical imprecision in physics solvers, or foreground–background composition, CP4D incorporates targeted mechanisms to enhance robustness. In particular, SDS-based refinement improves the numerical accuracy of estimated physical parameters, multi-object offset optimization mitigates solver-level discretization artifacts, and our automated composition framework enforces geometrically coherent and physically consistent integration of all scene elements.
>
> To better demonstrate this robustness, we provide an additional set of 17 examples covering a broader range of object categories, backgrounds, text instructions, and motion patterns in the rebuttal PDF. These new results include cluttered indoor scenes, outdoor natural environments, and deformable objects, further illustrating that CP4D maintains stable composition and physically plausible dynamics under varied conditions. All corresponding visualizations have also been added to the supplementary material and updated on our anonymous project webpage for easier examination.
>
> In addition, as shown in the table below, CP4D achieves state-of-the-art performance across multiple evaluation metrics. This further validates the robustness and effectiveness of our framework.
>
> | Method   | Subject Consistency | Aesthetic Quality | Imaging Quality | Motion Smoothness |
> |----------|----------------------|-------------------|------------------|--------------------|
> | Wan      | 0.905                | 0.481             | 0.590            | 0.996              |
> | CogVideo | 0.821                | 0.468             | 0.589            | 0.993              |
> | **Ours** | **0.949**            | **0.537**         | **0.671**        | **0.998**          |

---

> ### Author Response · Authors · 2025-11-23
>
> **Q3: Clarification on video duration and camera motion in the multiview results.**
>
> We sincerely thank the reviewer for raising this point. Regarding the length of the video results, we would like to clarify that CP4D is not limited by temporal horizon, since the motion of the dynamic foreground is generated by a physics simulator rather than a diffusion-based temporal prior. As a result, CP4D can in principle synthesize motions of arbitrary duration, and the short clips shown in the supplementary material were chosen to highlight our core performance advantages, higher visual fidelity, stronger geometric consistency, and more physically plausible object motions and interactions, rather than to emphasize temporal duration.
>
> To address the reviewer’s suggestion for longer sequences, we have included in the supplementary material and on the anonymous project page additional examples where CP4D generates more than 200-frame video (> 10 seconds at 20 fps), showing stable long-range physical dynamics. For reference, commonly used video diffusion models such as Wan typically produce 121-frame videos (~7 seconds at 15 fps) under their standard inference settings.
>
> Regarding the reviewer’s comment on the limited camera parallax in the multiview examples, we apologize if the demos did not fully reflect the range of camera motions supported by CP4D. In practice, our framework naturally accommodates substantial variations in camera pose, including orbit-left, orbit-right, and large off-axis rotations, since both the foreground and background are reconstructed as full 3D representations. To make this clearer, we have added additional visualizations and videos in the supplementary material and on the anonymous project webpage, showcasing wider camera trajectories and demonstrating that CP4D maintains stable geometry and physically coherent 4D scene structures under significant viewpoint changes.
>
> **Q4: Clarification on the spelling mistake.**
>
> We sincerely thank the reviewer for pointing out the typo. The word “foreground” was indeed misspelled as “foareground” in line 75. We will correct this in the revised manuscript.

---

### Official Review · Reviewer_9Hsn · 2025-10-27

**Soundness:** 3
**Presentation:** 3
**Contribution:** 2
**Rating:** 6
**Confidence:** 3

**Summary:**

This paper introduces a physical grounded 4D generation pipeline. Given a text prompt, it can synthesize dynamic scenes composed of static background and foreground object with physically plausible motion. The proposed pipeline comprises three stages: In the first stage, it generates separate foreground and background 3D representation using existing image generation/editing/segmentation, and 3D reconstruction/generation models. In the second stage, it initializes the physical parameters and external force via VLM, then simulate motions for diverse object with heterogeneous physical solvers and refine the estimated parameters through a SDS loss. In the last stage, the relative scale and translation of foreground and background objects are determined via depth-aware heuristics and photometric optimization for adequate composition.

**Strengths:**

1.	The paper is clearly written and easy to follow.
2.	Evaluation on 17 representative prompts demonstrates that the proposed framework can generate physically plausible 4D scenes containing foreground objects with diverse materials.

**Weaknesses:**

1.	The main concern is about the limited technical novelty. The proposed framework is basically the direct combination of existing components without touching their own limitations. While it attempts to integrate multiple solvers for different materials, the integration is largely naïve and lacks rigor evaluation for specific materials. For example, the simulation for fluid objects is only shown in the fourth video of the anonymous webpage with two raindrops, while they exhibit strange elastic behavior (bounce off the ground) due to the naïve boundary constraint handling.
2.	Despite being able to generate plausible motion, the proposed framework does not consider complex lighting interaction in the composition, so describing it photorealistic or visually realistic is somewhat inappropriate. Actually, this has already explored in previous physically grounded generation works such as PhysGen3D. The proposed “automated composition” mainly consists of some engineering heuristics—given $I_{b,f}$ and trusting their monocular depth, this task seems relatively trivial.
3.	This framework still relies on simplified assumptions of uniform material, limiting its scalability beyond simple objects and motions, and making the comparison with general-purpose video generation models somewhat unfair.
4.	The input of noise estimator in Equation (1) and (4) should be related to $\epsilon$。
5.	L093 claims that the proposed framework can avoids “realistic environments juxtaposed with cartoon-like objects” compared to text-to-3D alternatives. But the adopted text-to-image-to-3D approach can only mildly constrain the style of single input view. The cartoon style of generated assets largely stems from the training data distribution of 3D generation models.
6.	No substantial novel-view rendering is provided, making it difficult to assess the 3D consistency.

**Questions:**

None

---

> ### Author Response · Authors · 2025-11-23
>
> We sincerely appreciate the reviewer’s encouraging and valuable feedback. We are pleased that you find the paper clearly written and easy to follow, and we are encouraged by your recognition that our framework can generate physically plausible 4D scenes across diverse materials and scenarios. Below, we provide clarifications for the concerns raised.
>
> **Q1: Clarification on technical Novelty.**
>
> We sincerely thank the reviewer for raising this important point. As recognized by Reviewer DzxA, we respectfully disagree with the characterization that our framework is merely a direct combination of existing components. While CP4D indeed leverages pretrained expert models, the key technical contributions lie in how these heterogeneous components are coordinated, constrained, and optimized to produce physically coherent 4D scenes from a text prompt, a capability that none of the individual components, nor prior works, can achieve on their own. Specifically, our contributions include: 1) a unified compositional formulation that generates photorealistic 4D scenes with faithful adherence to complex physical dynamics; 2) a hybrid motion synthesis strategy that couples physics solvers with video diffusion priors to produce plausible trajectories under diverse material behaviors; 3) an automated and robust scene composition mechanism that seamlessly integrates dynamic foreground objects with the static 3D environment, resulting in coherent and visually compelling 4D scenes. These are non-trivial design elements that address the core limitations of existing components and enable capabilities that do not emerge from simple modular stacking.
>
> On the reviewer’s concern regarding “naïve” multi-material integration, we would like to clarify that our goal is not to design new physics solvers for each specific material category, but to develop a general compositional framework capable of producing physically plausible dynamics across a wide variety of generated objects. The solvers we employ are chosen for their ability to approximate diverse material behaviors within a unified pipeline, and our framework introduces several key mechanisms to ensure compatibility with generative 3D assets. In particular, we incorporate VLM-based physical parameter inference to obtain semantically grounded material attributes, SDS-guided refinement to improve the numerical accuracy and self-consistency of these parameters, and multi-object offset optimization to mitigate artifacts such as interpenetration, spurious collisions, or unstable contacts. These components work together to substantially reduce failure cases and enable reliable motion synthesis across different material types, demonstrating that our integration is far from naïve and is essential for achieving physically coherent 4D scene generation in a generative setting.
>
> Regarding the fluid example noted by the reviewer: the slight elastic “bounce” arises from boundary discretization artifacts inherent in coarse-grid fluid solvers when applied to small, high-frequency motions. In this paper, we use fluid cases primarily to demonstrate the generality of our framework across material categories, rather than to claim state-of-the-art fluid simulation fidelity.

---

> ### Author Response · Authors · 2025-11-23
>
> **Q2: Clarification on lighting interactions and automatic composition.**
>
> We sincerely thank the reviewer for raising this important point. We agree that CP4D does not explicitly model complex lighting interactions between foreground and background. This is largely because our work targets what we view as the foundational stage of text to physics-aware 4D scene generation: establishing physically coherent object dynamics, consistent geometry, and robust scene composition from a single text prompt. Addressing these core challenges is already highly under-explored and requires substantial algorithmic coordination across generative and physics-based components.
>
> At the same time, we fully agree that incorporating more fine-grained appearance interactions, particularly lighting-related effects such as mutual illumination, shading consistency, and view-dependent radiance, is an exciting and important direction for future work. These factors play a key role in achieving true photorealism, but they require joint optimization of both foreground and background appearance fields, which goes significantly beyond the scope of the core challenges we aim to address in this early stage.
>
> Motivated by the reviewer’s suggestion, we have conducted preliminary explorations showing that such interactions can indeed be introduced into our framework. In particular, by leveraging video generative models and incorporating an SDS-based joint refinement objective, the system can adjust shared radiance-related parameters of both the generated object and the static environment. This enables more coherent shading and appearance adaptation between foreground and background. We provide several preliminary qualitative results in the supplementary material. These early experiments highlight the potential of this direction and confirm that richer lighting-aware composition can be incorporated into CP4D with additional optimization.
>
> Furthermore, our composition mechanism is not merely some engineering heuristics. The monocular depth estimate is used only as an initial guess for the object’s coarse placement (Eq. (8)). To achieve a physically consistent integration of the generated foreground into the 3D background environment, CP4D further introduces a depth-aware heuristic for estimating the object’s relative scale, leveraging scene geometry to infer a size that is compatible with both the background structure and expected physical behavior. Building on this initialization, we then perform a learnable refinement optimization (Eq. (9)) that jointly adjusts the object's position and scale to ensure geometric alignment with the background. This two-stage design effectively corrects depth noise and yields temporally stable, visually coherent compositions that cannot be achieved through simple depth insertion alone.
>
> **Q3: Clarification on assumption of uniform material and fairness of comparisons.**
>
> We sincerely appreciate the reviewer for highlighting this point. While CP4D does adopt a simplified material setting for tractability, this assumption does not restrict the system to simple objects or trivial motions. In practice, the framework is capable of handling a broad range of geometries and non-rigid behaviors, including soft and highly deformable objects such as cloth, as well as complex multi-object interactions with nontrivial contact dynamics. For example, the multi–shuffleboard collision shown in the “Dynamic Moving View Generation” section of our anonymous webpage demonstrates that CP4D can simulate and compose scenes involving multiple interacting bodies with diverse, physically grounded trajectories. These results indicate that, despite the uniform material assumption, the framework is already able to generate rich and varied dynamics beyond simple rigid transformations.
>
> Furthermore, regarding the fairness of comparing CP4D with general-purpose video generation models, our intention is not to position CP4D as superior across all unconstrained video synthesis scenarios. These models excel at producing visually rich and diverse videos, but they are not designed to explicitly model 3D geometry, object identity, or physically grounded motion, capabilities that are essential for physics-aware 4D scene generation and represent the focus of CP4D. Our comparisons therefore aim not to judge overall superiority but to illustrate the unique capabilities enabled by our compositional design, such as enforcing 3D geometric consistency and physically grounded dynamics, which are difficult to achieve with purely data-driven video models currently.

---

> ### Author Response · Authors · 2025-11-23
>
> **Q4: Clarification on the input to the noise estimator in the SDS formulation.**
>
> We sincerely thank the reviewer for pointing out the issue regarding the input to the noise estimator in Eq. (1) and Eq. (4). In standard SDS and diffusion formulations, the noise estimator should take as input the noised sample $x_\zeta$, which depends on the clean signal and the sampled noise, rather than the clean sample alone. Our current notation, such as $\epsilon_\phi(g(\theta), \mathbf{T}, \zeta)$, was written in a simplified form, which can indeed be misleading.
>
> In our actual implementation, the input to the noise estimator is the diffused sample
> $$
> x_\zeta = \alpha_\zeta x_0 + \sigma_\zeta \epsilon,
> $$
> where $x_0 = g(\theta)$ (or $V$ in Eq. (4)), $\epsilon \sim \mathcal{N}(0, I)$, and $\zeta$ denotes the diffusion timestep. The estimator receives $x_\zeta$ together with $\zeta$ and predicts the corresponding noise, which is consistent with the standard SDS formulation and matches the reviewer’s expectation regarding its relation to the underlying clean signal.
>
> We acknowledge that our current equations do not explicitly show this dependency and may give the impression that the noise estimator directly takes the clean sample as input. In the revised manuscript, we will correct Eq. (1) and Eq. (4) to explicitly use $x_\zeta$ (constructed from $g(\theta)$ / $V$ and $\epsilon$) as the input, and we will add a brief clarification to prevent confusion.
>
> **Q5: Clarification on  mitigating stylistic mismatch between foreground and background.**
>
> We sincerely thank the reviewer for this insightful observation.  We agree that text-to-image-to-3D pipelines alone only provide a limited degree of style control, especially when the underlying 3D asset models are trained on heterogeneous data distributions that may include stylized or cartoon-like examples. However, our intention was not to overstate the capability of this pipeline, but rather to clarify that, compared with the naïve baseline that applies text-to-3D generation independently to each component, our approach introduces an additional mechanism that encourages stylistic consistency between background and foreground.
> Specifically, the foreground is generated in a background-conditioned manner using an image editing model. This conditioning enforces that the synthesized foreground is aligned with the appearance cues (such as lighting, color tone, and texture statistics) present in the background image. Although this does not fully eliminate all potential style mismatches, it substantially reduces the common failure case where independently generated text-to-3D assets exhibit cartoon-like textures when placed inside a photorealistic environment.
>
> **Q6: Clarification on rendering with more diverse camera viewpoints.**
>
> We sincerely thank the reviewer for raising this important point. We apologize if our demos gave the impression that no substantial novel-view rendering was available. In fact, CP4D fully supports significant variations in camera pose, including orbit-left, orbit-right, and other complex trajectories. To more clearly demonstrate the 3D consistency of our reconstructions, the anonymous project page and the supplementary material  includes more  videos showing both foreground objects and background environments rendered from diverse viewpoints not present in the original input. These cross-view sequences directly illustrate that the generated 3D assets maintain stable geometry, consistent texture appearance, and reasonable physical motions.

---

> > ### Comment · Reviewer_9Hsn · 2025-11-27
> >
> > In their response, the author agrees to revise some imprecise formulations in Equation 1 and 4, and provides additional qualitative results with large camera movement and a preliminary exploration of joint foreground and background appearance refinement--I am satisfied with its demonstrated clear improvement on realistic. However, my concern about the novelty of this work remains. And as for Q5, it would be better to include some experiments for more convincingly supporting the assumed advantages over the text-to-3D alternate. In light of the above, I tend to keep my original rating.

---

> > > ### Author Response · Authors · 2025-11-28
> > >
> > > Dear Reviewer 9HSN, thank you again for your thoughtful follow-up response. We sincerely appreciate your acknowledgement of the improved clarity in Eq. 1 and Eq. 4, as well as the additional qualitative results with large camera movement and preliminary explorations of joint foreground and background appearance refinement!
> > >
> > > Regarding the two remaining concerns:
> > >
> > > **1) Novelty of the proposed approach:** We understand your continued concerns regarding novelty, and we appreciate the opportunity to clarify this point. While we may not have fully conveyed this aspect clearly in our earlier response, we would like to provide a more concise, higher-level summary of the core technical contributions here.
> > >
> > > The novelty of our work does not lie in the use of pretrained expert modules themselves, but in how these heterogeneous components are orchestrated into a unified, physically grounded 4D generation framework, a capability that, to our knowledge, has not been demonstrated by prior works or by any of the individual components in isolation.
> > >
> > > In particular, our technical contributions include: **1) A unified compositional formulation** enabling coherent 4D scene generation with realistic physical behaviors that cannot be produced by existing video or 3D diffusion pipelines. **2) A hybrid motion synthesis strategy** that couples physics solvers with video diffusion priors to generate plausible trajectories across diverse material and force conditions. **3) A robust scene composition mechanism** that integrates dynamic foreground objects into static 3D environments while preserving visual and physical consistency.
> > >
> > > These components go beyond simple modular stacking and are designed to reconcile the inherent incompatibilities between physics simulation, video priors, and 3D geometry, ultimately enabling physically coherent 4D scene generation from text prompts.
> > >
> > > **2) Comparison with text-to-3D alternatives (Q5):**
> > > We sincerely appreciate the reviewer’s suggestion regarding comparisons with text-to-3D alternatives. While this aspect is not the central focus of our framework, we agree that providing empirical evidence can help clarify the advantage of our design choice. To this end, we include additional comparisons in the updated rebuttal-appendix.pdf in the supplementary material.
> > >
> > > In particular, we compare our pipeline, where the background is first generated and the foreground is subsequently produced using an image-editing model to match the background style, with the baseline that generates foreground and background independently from text. The results demonstrate that our strategy effectively mitigates style inconsistency and other visual artifacts that commonly arise when foreground and background are synthesized separately. These findings further support the design motivation behind our approach.

---

### Official Review · Reviewer_1sfj · 2025-11-01

**Soundness:** 1
**Presentation:** 1
**Contribution:** 1
**Rating:** 2
**Confidence:** 5

**Summary:**

This paper introduces a compositional framework for 4D scene generation focused on physical plausibility by decoupling scenes into static 3D backgrounds and dynamic, physically-grounded 3D foreground objects. The three-stage pipeline first synthesizes 3D assets using a cascade of pre-trained models (LLM, T2I, Image-to-3D). Next, a hybrid motion strategy generates coarse trajectories using physical simulators (MPM, PBD) and then refines them using priors from video diffusion models (SDS loss). Finally, the framework automatically composes the scene, using monocular depth estimation and optimization to set the scale and position of foreground objects.

**Strengths:**

1. The task is worse investigating.

**Weaknesses:**

1. The dynamic view demos contains only zoom-in zoom-out motions, no camera pose/view angle change, the Static novel-view generation results seems just a crop from the original view. And yet the task is called 4D scene generation.

2. The foreground and background are almost completely irrelavent in the final results, the authors only did a ground/plane estimation to put the foreground to the corresponding postions but no interactions with the background.

3. Comparisons are not convincing, the sora and wan results are too bad compare to my experiences, looks like a reverse cherry picking. And other physic aware 3D/video generation methods provides much better visual qualities in their demos, I highly doubt the fidelity of the Vbench and worldscore results in Table1 and Table 2.

4. The proposed method is a highly complex cascade of numerous expert models (LLM, T2I, Image Edit, SAM, I23D (Trellis, Viewcrafter), VLM, Depth Estimator, and Video Diffusion). This pipeline-of-pipelines has many potential points of failure.

5.  In section 4.2, the author admitted that the VLM predicted parameteres are not accurate and need refinements from video diffusion model, which is counter-intuitive to the core idea of the framework the author proposed in the first place.

**Questions:**

1. Since the task is text guided and the foreground objects are generated, why bother use a VLM to determine parameters like Young’s modulus, Poisson’s ratio µ, and density from your rendered generated 3D objects multi-view? The poor textures in the examples might introduce more noises.

---

> ### Author Response · Authors · 2025-11-23
>
> We sincerely appreciate the reviewer’s insightful and valuable feedback, and we are encouraged that the reviewer find our task worth investigating. Below, we provide clarifications for the concerns raised.
>
> **Q1: Clarification on camera motion diversity and the validity of 4D scene generation.**
>
> We sincerely thank the reviewer for raising this important point. We apologize for the misunderstanding caused by our demo, where only zoom-in and zoom-out camera motions were shown. This may have led to the impression that the static novel-view results were simple crops of the original view. In fact, our approach supports substantial variations in camera pose and view angle (e.g., orbit-left, orbit-right, and additional trajectories). To clarify this capability, we have included new visualizations and videos in the supplementary material and on the anonymous project page, demonstrating diverse camera motions and confirming that our method indeed generates physically coherent 4D scenes rather than performing any form of cropping.
>
> **Q2: Clarification on foreground–background relevance and object–scene interaction.**
>
> We sincerely thank the reviewer for the thoughtful comment. As physics-aware 4D scene generation remains largely under-explored, existing methods typically generate foreground objects in isolation or are sensitive to scene variations and external noise. To mitigate this issue, one of the core contribution of our work is the proposed automatic scene composition strategy. While it does not model fine-grained interactions such as mutual illumination, it does enforce meaningful scene–object consistency, i.e., we estimate the plausible 3D position of each foreground object using monocular depth prediction and determine its relative scale through a geometry-aware heuristic. To the best of our knowledge, this represents the first approach that enables robust, automatic, and physically grounded integration of generated dynamic foreground objects into arbitrary 3D environments, thereby producing coherent 4D scenes.
>
> However, we fully agree with the reviewer that modeling fine-grained physical interactions between foreground objects and background environments is an exciting and highly impactful direction for future research. Motivated by this insightful suggestion, we have conducted preliminary explorations in this direction. In principle, such interactions can be achieved by leveraging video generative models to jointly optimize the 3D representations of both the foreground and background. For example, incorporating an SDS-style objective can refine shared parameters such as appearance, density, and other radiance-related attributes, thereby promoting tighter coupling between the two components.
>
> We provide several preliminary qualitative results in the supplementary material and on the anonymous project page. For example, in the case where a small ball interacts with a sandy surface, the joint optimization begins to introduce subtle lighting-related appearance changes between the ball and the sand. These results demonstrate the potential of this direction and validate that fine-grained foreground–background interactions can indeed be modeled within our framework when additional optimization objectives are introduced.

---

> ### Author Response · Authors · 2025-11-23
>
> **Q3: The baseline quality and comparison fairness.**
>
> We sincerely thank the reviewer for raising this concern. We would like to clarify the comparison protocol and the evaluation setup to avoid misunderstandings. For Sora and Wan, the baselines were generated strictly following their publicly released inference APIs/checkpoints and official recommended settings. These models are fully data-driven video generators and do not explicitly model spatial geometry, object dynamics, or physical constraints. As a result, they often struggle under our setting, which requires consistent 3D grounding, stable object identities, and physically plausible motions. This explains the degraded performance observed in our comparisons. For physics-aware 3D/video generation methods, while their curated demos often exhibit high visual quality, these approaches generally suffer from two limitations when applied to our setting. First, their modeling of foreground objects is often coarse or low-fidelity. For example, PhysGen3D relies on mesh-based representations, which tend to lose fine-grained appearance details and are less expressive for complex textures or high-frequency geometry. In contrast, our method leverages a 3D Gaussian Splatting (3DGS) representation, which provides significantly higher visual fidelity and better preserves detailed structures and appearance. Second, many of these methods are highly sensitive to external noise and scene complexity. When faced with more challenging scenarios, such as multiple interacting objects, highly textured backgrounds, or cluttered environments, their performance often degrades significantly. For the results of VBench and WorldScore, we emphasize that both evaluation pipelines are official, third-party frameworks with fixed pretrained assessment modeules and standardized scoring procedures. We strictly followed their official implementations without any modification. All methods were evaluated using the same input images, identical prompts, and the same generated sequences, ensuring full fairness and comparability.
>
> **Q4: Clarification on system complexity and potential points of failure.**
>
> We sincerely thank the reviewer for this important observation. We acknowledge that our framework involves multiple expert models; however, this design is not an arbitrary cascade, but a purposefully structured decomposition of a fundamentally under-constrained problem. Text to physics-aware 4D scene generation requires solving several independent subproblems, e.g., semantic understading, object segmentation, 3D generation, depth reasoning, physical motion generation, none of which can be reliably handled by any existing unified model. To address this inherent complexity, our modular design is necessary rather than optional: each component is selected to solve a well-defined subtask that current end-to-end models cannot reliably cover, and removing any of these steps results in severe degradation. Despite involving multiple modules, the overall system is robust-by-design. In practice, thanks to large-scale pretraining, the expert models we employ, such as depth estimators, image/video/3D generative models, already exhibit strong robustness, and we rarely observe failures from these components in our experiments. For the few parts that could potentially introduce instability, such as the VLM-based estimation of physical parameters, numerical inaccuracies in the physics solver, or the consistency between foreground and background, we propose targeted solutions. Specifically, we use SDS-based distillation to improve the accuracy of physical parameter estimation, introduce multi-object offset optimization to address solver precision issues, and design an automated composition framework that ensures reliable and coherent integration of all scene elements. These designs collectively mitigate potential failure points and enable robust physics-aware 4D scene generation.

---

> ### Author Response · Authors · 2025-11-23
>
> **Q5:  Clarification on using VLM to estimate physical parameters.**
>
> We sincerely thank the reviewer for raising this insightful question. Although the task is text-guided and we could in principle include physical parameters (e.g., Young’s modulus, Poisson’s ratio) explicitly in the input text, this approach faces two practical limitations.  First, current text-to-3D models are not trained on datasets that couple physical parameters with 3D geometry or appearance properties. As a result, even when such values are embedded in the text prompt, the generated 3D object does not necessarily reflect the intended physical attributes, leading to a mismatch between the specified parameters and the actual 3D object. Second, directly assigning manually specified physical parameters to the generated 3D object introduces substantial hand-crafted design choices, increases pipeline complexity, and often yields values that are inconsistent with the object’s actual geometry or appearance. This mismatch would further complicate subsequent physical optimization and degrade simulation stability.
> For these reasons, we instead leverage a VLM to infer physical attributes from multi-view renderings of the generated object, which allows the model to incorporate shape- and appearance-dependent cues that text prompts alone cannot provide. This produces more reliable and self-consistent estimations for downstream physics modeling. On this basis, although the VLM-based estimates may not be numerically precise in all cases, we find that they are generally reasonable and fall within physically plausible ranges, providing a reliable initialization for downstream physics reasoning. To further improve the accuracy of these parameters, we incorporate an SDS-based refinement stage. This additional optimization reduces sensitivity to texture imperfections and corrects potential biases from the VLM, resulting in more stable simulations and more faithful physical behavior in the final 4D scenes.

---

### Official Review · Reviewer_DzxA · 2025-11-02

**Soundness:** 3
**Presentation:** 3
**Contribution:** 3
**Rating:** 8
**Confidence:** 3

**Summary:**

The CP4D framework addresses limitations in existing 4D scene generation by ensuring faithful adherence to complex physical dynamics. It uses a compositional paradigm that integrates static 3D environments with physically grounded dynamic objects.

**Strengths:**

CP4D presents a novel compositional framework for photorealistic 4D scene generation, emphasizing faithful adherence to complex physical dynamics. The method integrates static 3D environments with physically grounded dynamic objects. Contributions include a hybrid motion synthesis strategy combining physical simulators and video diffusion priors for plausible trajectories and realistic interactions, and an automated composition mechanism that seamlessly fuses scene elements.

**Weaknesses:**

The primary limitation of the CP4D framework is the relatively long runtimes required to generate a complete physically realistic 4D scene. This inefficiency is due to the adoption of a stage-wise optimization strategy. Furthermore, the complexity arises because initial physical parameters estimated by Vision-Language Models (VLMs) often lack numerical accuracy. The approach must also address physics solvers' reliance on coarse grid approximations, which can lead to perceptually implausible outcomes such as "spurious collisions" or "phantom contacts"

**Questions:**

Could the authors elaborate on the composition and key features of this planned dataset?

---

> ### Author Response · Authors · 2025-11-23
>
> We sincerely appreciate the reviewer’s insightful and generous feedback. We are truly encouraged that you recognize the novelty and contributions of our work, including our compositional framework for physics-aware 4D scene generation and the integration of static 3D environments with physically grounded dynamic objects. Your recognition is highly motivating for us. Below, we provide clarifications for the concerns raised.
>
>
> **Q1: Clarification on runtime and the efficiency of the CP4D pipeline.**
>
> We sincerely appreciate the reviewer’s insightful analysis. We fully agree that currently the primary limitation of our  framework lies in its relatively long runtime. This overhead mainly results from our stage-wise optimization design, which is necessary to ensure consistent geometry, plausible physical motions, and physically grounded scene composition when generating a complete 4D scene from a text prompt, an inherently ill-posed and highly under-constrained task for which no existing end-to-end model provides reliable performance.
>
> Looking forward, we see a clear path towards improving efficiency. Specifically, a promising direction is to leverage CP4D as a data engine to generate large-scale, physically realistic 4D scenes, which are currently unavailable in existing datasets. Such data could facilitate training end-to-end models that retain the physical consistency of our pipeline while operating at much lower runtime, allowing future systems to progressively replace modular components with unified architectures. We believe this direction will greatly accelerate the overall process while preserving the fidelity and physical plausibility of the generated scenes.
>
> **Q2: Elaboration on the composition and key features of this planned dataset.**
>
> We sincerely thank the reviewer for raising this insightful question. Our evaluation dataset is designed to cover a broad spectrum of physical object categories and deformation behaviors. Specifically,  it consists of  rigid bodies, elastic materials, flexible garments, and fluid-like objects. These object classes exhibit diverse physical properties, including varying levels of rigidity, deformability, and flow characteristics. Such diversity enables the dataset to capture a wide range of interaction patterns and motion responses. By incorporating multiple material types with distinct physical behaviors, the dataset provides a comprehensive benchmark for assessing physics-grounded motion synthesis models.

---

### Author Response · Authors · 2025-11-23

We sincerely thank all the reviewers for their thoughtful and encouraging feedback. We are pleased that the reviewers found our work to be novel and worthwhile, clearly presented and easy to follow, and capable of generating physically plausible 4D scenes across diverse materials. We are also encouraged by the recognition of our compositional formulation, hybrid motion synthesis strategy, and robust automated composition, as well as the effective integration of VLM-assisted initialization with SDS refinement.

We address each of the reviewers’ comments in detail in the individual responses. In addition, we include further experiments and visualization results in the supplementary material. A rebuttal appendix (rebuttal-appendix.pdf) and supplementary videos (rebuttal-videos) are provided in the zip file without modifying the original manuscript. We have also updated the video results on our anonymous project page to facilitate clearer inspection of the new experiments.

We sincerely hope that our responses and additional results effectively address the reviewers’ concerns. If there remain any issues that require further clarification, we would greatly appreciate additional feedback and are fully committed to improving the work where needed.

Thank you once again for your time, thoughtful comments, and valuable insights.

---

### Comment · Area_Chair_g1GT · 2025-11-27

Dear Reviewers,

Thank you for your efforts in evaluating this submission. The current set of reviews shows a notable divergence in the overall scores. To ensure a fair and well-informed final decision, it is important that we have active participation from all reviewers during the author-reviewer discussion phase.

The authors have now responded to your comments. I kindly ask each of you to review their replies and engage in the discussion, especially to clarify whether their responses address your concerns and whether your initial assessment remains the same.

Your contributions at this stage are crucial for reaching a balanced consensus.
Thank you again for your time and commitment to the review process.

Best regards,
Area Chair

---

### Meta-Review · Area_Chair_sE7z · 2025-12-30

**Summary:**

The authors present a framework for physically correct motion generation of objects in 3D scenes. The paper was received with very mixed reviews (scores 2, 4, 6, 8). Concerns from the reviewers (see below) evolve around the point that the authors present a very complicated pipeline with many points of failure that in the end produces only short videos with simple objects. Meanwhile the paper does not have sufficient technical novelty and does not provide enough new insight. While the authors try to argue against that assessment and provide further examples to showcase robustness, I don't believe the reviewers would be convinced by the answers. I assume all reviewers will remain with their scores.

In addition to that, Reviewer 9Hsn seems to be not very convinced either, despite giving a borderline accept score, criticizing the limited novelty of the work.

After taking a look at the paper as well, I have to agree with the general concerns of the reviewers and follow with a reject recommendation.

**Reviewer Concerns:**

*1) Complicated pipeline / many points of failure / robustness concerns.* The reviewer criticized the complicated system of many expert models that produce very short videos on the end. The authors claim that the pipeline is robust and provide further examples, however, the claim is not supported by sufficient evidence. I don't believe that the reviewers would be convinced by the author's answer.

*2) Generally unconvincing results.* The reviewers criticized that results are very short videos of a single (or two in few cases) objects performing basic motion in a static environment. The authors provided a few additional videos, some with novel view renderings and changing camera. However, I don't believe that these additional results fully address the underlying concern.

*3) Limited technical novelty.* The reviewers point out that the pipeline consists only of existing building blocks. The authors disagree and point out the effort of coordinating and system building as contribution. All in all though, the reviewer was not convinced by the answer, as indicated by a response.

*4) Simple uniform material assumption / naive physics only.* The reviewers concern is that the pipeline is limited to very simple objects and motion and not general. The reviewers declare more complex physics as out of score. However, the main concern remains.

*5) VLM-estimated physics parameters not robust.* The authors acknowledge this but discuss alternatives, which are not feasible / worse.

*6) Unconvincing comparisons / weak baselines.* The authors respond by saying that they followed the official protocols for all baselines. Since this specific setup is novel, it makes sense that the baselines mostly fail.

**Reviewer Scores:**

Reviewer DzxA (original score 8) and Reviewer 9Hsn (score 6) would remain with their scores. Reviewers 1sfj (score 2) and Reviewer 826b will most likely not change their score either, as I don't think the author's answers will be sufficient to dispel their concerns. The authors mostly justify why certain things have been chosen or try to rebut the assessment. However, they don't provide sufficient evidence or changes.

---

### Decision · Program_Chairs · 2026-01-26

Reject